# CoMind: Towards Community-Driven Agents for Machine Learning Engineering

**Sijie Li**[1*]   **Weiwei Sun**[2*]   **Shanda Li**[2]   **Ameet Talwalkar**[2,3]   **Yiming Yang**[2]

[1]Peking University   [2]Carnegie Mellon University   [3]Datadog
planarg@stu.pku.edu.cn

## Abstract

Large language model (LLM) agents show promise in automating machine learning (ML) engineering. However, existing agents typically operate in isolation on a given research problem, without engaging with the broader research community, where human researchers often gain insights and contribute by sharing knowledge. To bridge this gap, we introduce MLE-Live, a live evaluation framework designed to assess an agent's ability to communicate with and leverage collective knowledge from a simulated Kaggle research community. Building on this framework, we propose CoMind, a multi-agent system designed to systematically leverage external knowledge. CoMind employs an iterative parallel exploration mechanism, developing multiple solutions simultaneously to balance exploratory breadth with implementation depth. On 75 past Kaggle competitions within our MLE-Live framework, CoMind achieves a 36% medal rate, establishing a new state of the art. Critically, when deployed in eight live, ongoing competitions, CoMind outperforms 92.6% of human competitors on average, placing in the top 5% on three official leaderboards and the top 1% on one.

## 1 Introduction

The capabilities of large language model (LLM)-based agents are rapidly advancing, showing significant promise in automating complex tasks across domains including software engineering (Jimenez et al., 2024; Merrill et al., 2026), mathematical problem-solving (Ren et al., 2025; Wu et al., 2025; Collins et al., 2025; Li et al., 2026), and scientific discovery (Romera-Paredes et al., 2024; Yamada et al., 2025; Sun et al., 2025; Feng et al., 2026). A particularly challenging and impactful frontier for these agents is machine learning engineering (MLE). Automating the multifaceted MLE pipeline, which spans the design, implementation, and rigorous evaluation of high-performance models, remains a critical test of an agent's autonomous reasoning and decision-making abilities.

Recent advances have introduced LLM agents capable of autonomously developing machine learning pipelines for Kaggle-style competitions (Chan et al., 2025). Current approaches have demonstrated a range of techniques, from the ReAct-style reasoning in MLAB (Huang et al., 2024) and the tree-based exploration of AIDE (Jiang et al., 2025), to the skill-specialized multi-agent system of AutoKaggle (Li et al., 2024). Although these systems represent important steps toward automating MLE, they are fundamentally designed to operate in isolation, exploring the solution space individually.

This isolated approach stands in stark contrast to how human experts operate. In real-world data science competitions and research, participants thrive on community knowledge sharing: learning from public discussions, shared code, and collective insights to enhance solution quality and drive innovation (Wuchty et al., 2007). By failing to engage with this dynamic external context, current agents are prone to converging on repetitive strategies and ultimately plateauing in performance. This critical gap motivates our central research question:

*How can we **evaluate** and **design** research agents that utilize collective knowledge?*

---

*Equal contribution.

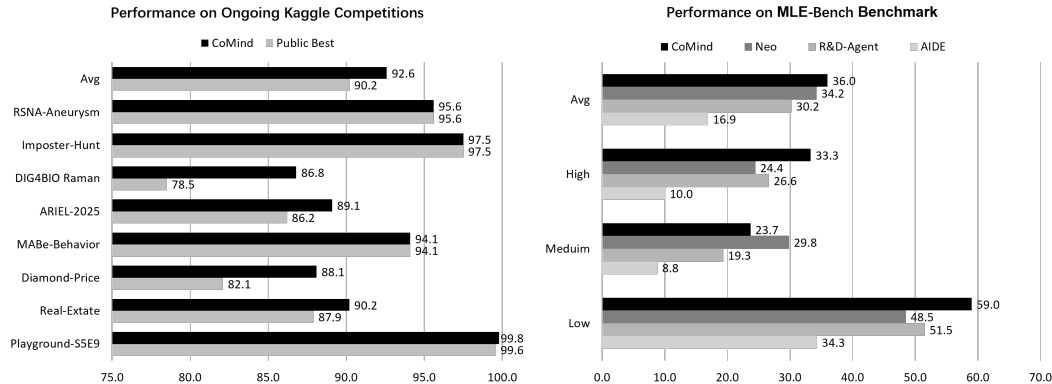

Figure 1: **Left:** CoMind's win rates on eight ongoing Kaggle competitions compared with the public best entry. **Right:** *Any Medal* results on 75 MLE-Bench competitions grouped by task difficulty levels. CoMind achieves state-of-the-art performance on MLE-Bench compared to strong baselines.

To address this question, we introduce **MLE-Live**, a controllable evaluation framework that simulates realistic Kaggle-style research communities with time-stamped public discussions and shared code artifacts that public before competition deadline. This ensure the information access is same as human participant. MLE-Live enables rigorous evaluation of agents' ability to leverage community knowledge in temporally grounded settings, supporting both offline evaluation on past competitions and online evaluation on ongoing competitions.

Building upon this framework, we propose **CoMind**, a multi-agent system designed to systematically incorporate external knowledge and iteratively refine solutions. CoMind's architecture consists of five specialized agent role operating in concert. A central *Coordinator* manages the overall workflow and community interactions. To process external knowledge, an *Analyzer* first summarizes and suggests on improvements and weaknesses for a curated group of solutions, while an *Idea Proposer* brainstorms a diverse pool of ideas and synthesizes novel strategies. These strategies are then passed to multiple parallel *Coding Agents* for implementation and report generation. Finally, a dedicated *Evaluator*, which creates robust scripts for solution assessment and selection. This collaborative process allows CoMind to effectively utilize external community knowledge and construct novel solution for the targeted research problem.

We conducted a comprehensive, two-pronged evaluation to assess CoMind's performance in both static and live environments. First, on a static benchmark comprising 75 past Kaggle competitions from MLE-Bench (Chan et al., 2025), CoMind achieved an overall medal rate of 0.36, establishing a new state of the art by significantly outperforming prior leading agents such as Neo and ML-Master (Liu et al., 2025). Second, to validate its real-world practicality, we deployed CoMind in eight ongoing Kaggle competitions (detailed in Figure 1). In this challenging live setting, CoMind proved highly effective, achieving an average rank better than 92.6% of human competitors while placing in the top 5% on three official leaderboards and the top 1% on one. These results demonstrate CoMind's robust effectiveness against contemporary challengers.

In summary, our contributions are:

- **MLE-Live**: A live evaluation framework simulating community-driven machine learning research with realistic shared discussions and code.
- **CoMind**: A novel agent excelling at collective knowledge utilization and iterative exploration, achieving medal-level performance in real competitions.
- **Community-Driven Multiagent Collaboration**: An iterative parallel exploration mechanism enabling continuous knowledge accumulation.

## 2 RELATED WORK

The rise of large language models (LLMs) has sparked a new wave of research into LLM-driven agents, systems that leverage LLMs' reasoning and language capabilities to autonomously perceive,

plan, and act within digital or physical environments. Early works such as ReAct (Yao et al., 2023; Schick et al., 2023; Shen et al., 2023; Hong et al., 2023; Boiko et al., 2023) introduced frameworks that transform LLMs into programmable reasoning engines by interleaving natural language reasoning with tool-use actions. Subsequent studies have extended these agents to various domains, including computer usage (Xie et al., 2024; Zhou et al., 2024) and software development (Wang et al., 2025; Jimenez et al., 2024).

In parallel, the field of automated machine learning (AutoML) aims to reduce human involvement in building ML pipelines by automating tasks such as model selection, hyperparameter tuning, and architecture search. Early systems like Auto-WEKA (Thornton et al., 2013), HyperBand (Li et al., 2018) and Auto-sklearn (Feurer et al., 2022) used early stopping and Bayesian optimization to search over pipeline configurations, while methods like DARTS (Liu et al., 2019) expanded automation to neural architectures. More recent frameworks such as AutoGluon (Erickson et al., 2020) and FLAML (Wang et al., 2021) emphasize efficiency and ease of use.

Building on these developments, recent efforts have applied LLM-based agents to machine learning engineering (MLE) tasks (Hollmann et al., 2023; Guo et al., 2024; Li et al., 2024; Grosnit et al., 2024; Hong et al., 2024; Chi et al., 2024; Trirat et al., 2024; Huang et al., 2024). However, most evaluations remain constrained to closed-world settings with predefined search spaces, offering limited insight into how these agents perform in open-ended or collaborative ML environments. While some agents (Guo et al., 2024; AI-Researcher, 2025) incorporate basic retrieval tools, these are typically based on simple semantic matching, and robust evaluation methodologies remain underdeveloped.

Meanwhile, several benchmarks have been proposed to evaluate machine learning (ML) engineering capabilities. MLPerf (Mattson et al., 2020) assesses system-level performance, including training speed and energy efficiency. To evaluate end-to-end ML workflows, MLAB (Huang et al., 2024) tests the capabilities of LLM-based agents across 13 ML tasks. MLE-Bench (Chan et al., 2025) and DSBench (Jing et al., 2025) further extends to about 75 Kaggle competitions covering tasks such as preprocessing, modeling, and evaluation. However, these benchmarks typically evaluate agents in isolation, overlooking the collaborative dynamics of real-world ML development. In contrast, our work introduces a framework that simulates community-driven settings, enabling evaluation of agents' ability to engage with and benefit from shared knowledge, while ensuring that resource access remains fair and realistic.

## 3 MLE-LIVE

Existing machine learning benchmarks typically evaluate agents in static, isolated environments. This approach fails to capture the dynamic and collaborative nature of real-world platforms like Kaggle, where progress is driven by community knowledge sharing. Participants constantly learn from shared code, public discussions, and the iterative work of others, making these community interactions a decisive factor in developing top-tier solutions.

To bridge this gap, we introduce **MLE-Live**, a live evaluation framework that extends the widely-used MLE-Bench (Chan et al., 2025). The core innovation of MLE-Live is its simulation of community interactions, providing agents with a time-stamped stream of discussions and code artifacts that mirrors the natural flow of public knowledge during a competition.

Each competition environment in MLE-Live includes the following components: (i) Task description: The background, specifications, evaluation metrics, and data structure, scraped directly from the original Kaggle competition. (ii) Competition dataset: A cleaned train-test split of the official data. When necessary, this includes reconstructed test sets to account for data that is no longer public. (iii) Submission grader: An evaluation script that precisely mimics Kaggle's official scoring mechanism. (iv) Leaderboard: A snapshot of the final public leaderboard. (v) Community artifacts: A curated set of discussions and code notebooks that were **published before the competition deadline**. These artifacts are enriched with valuable metadata (e.g., vote counts, public scores, author tiers) to signal quality and are accompanied by any public datasets or models they reference, creating a self-contained and realistic research environment.

MLE-Live aggregates a substantial dataset of 12,951 discussions and 15,733 kernels from 75 Kaggle competitions. To ensure fairness and eliminate post-hoc data leakage, it strictly includes only resources available prior to competition deadlines, forcing agents to operate under the same information constraints as human participants. This approach offers numerous benefits for robust evaluation:

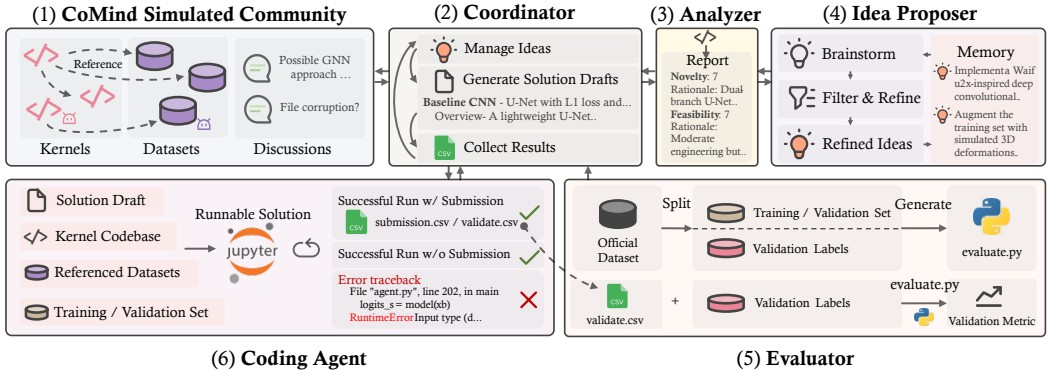

Figure 2: Overview of CoMind. Specialized agents (Coordinator, Analyzer, Idea Proposer, Coding Agent, Evaluator) interact with a simulated Kaggle community of kernels, datasets, and discussions.

it grounds agents in diverse, objectively-graded ML problems from Kaggle, while the controlled information scope allows for a fair assessment of their retrieval and reasoning abilities. These features enhance reproducibility and enable consistent, longitudinal comparisons between different agents.

## 4 CoMind

We propose **CoMind**, a community-augmented large language model (LLM) agent designed to automate machine learning (ML) engineering in an iterative, collaborative setting. Figure 2 is an overview of CoMind workflows.

### 4.1 COMMUNITY SIMULATION

CoMind's effectiveness stems from simulating the collaborative dynamics that drive breakthrough performance in competitive ML environments. Unlike isolated automated ML systems, CoMind replicates how Kaggle participants leverage community knowledge: drawing insights from discussions, adapting public notebooks and datasets, and contributing discoveries back to the collective knowledge pool.

The simulated community is represented as $(\mathcal{K}_t, \mathcal{D}_t, \mathcal{T}_t)$ at iteration $t$, where $\mathcal{K}_t$ contains all kernels with evaluation metrics, $\mathcal{D}_t$ includes published datasets and model checkpoints, and $\mathcal{T}_t$ captures the dependency relationships between resources. CoMind initializes a high-quality community $(\mathcal{K}_0, \mathcal{D}_0, \mathcal{T}_0)$ by fetching $k_{\text{kernel}}$ top-performing kernels and $k_{\text{discussion}}$ most popular discussions from Kaggle, along with all referenced datasets and models. The system constructs a dependency graph $\mathcal{T}_0 = (V, E)$ where vertices represent kernels or datasets and edges capture resource dependencies.

This dependency structure enables CoMind to systematically trace solution construction, identify influential artifacts, and prioritize resources that drive performance improvements. The graph facilitates intelligent ensemble strategies by combining complementary approaches while avoiding redundant components.

CoMind operates as an active community participant, iteratively analyzing promising kernels, generating novel solutions, conducting experiments, and contributing successful results back to the community. Each iteration produces new artifacts: enhanced kernels, augmented datasets, or ensemble checkpoints, that expand the community knowledge base with associated performance metrics.

Through this continuous cycle of exploration and contribution, CoMind simulates the collaborative dynamics of competitive ML development, where collective intelligence progressively advances performance frontiers at automated scale and speed.

### 4.2 MULTI-AGENT SYSTEM

CoMind orchestrates machine learning experimentation through a coordinated multi-agent system. Specialized agents collaborate in distinct roles, mirroring the division of expertise in human research

teams across ideation, implementation, and evaluation. The workflow is an iterative loop managed by the Coordinator, which delegates tasks to the other agents.

**Coordinator** The *Coordinator* serves as **CoMind**'s central orchestration hub. Its primary responsibilities are managing the workflow, interfacing with the community environment, and allocating resources. At the start of each iteration $t$, the *Coordinator* initiates the process by strategically sampling promising code notebooks (kernels) $\mathcal{K}'_t$ and relevant datasets $\mathcal{D}'_t$ from the community. This focused sampling directs the system's attention toward high-potential areas. After receiving refined ideas from the *Idea Proposer*, the *Coordinator* translates them into concrete solution drafts $\mathcal{S}_t$, which are comprehensive blueprints detailing model architecture, feature engineering, and training procedures. It then instantiates multiple *Coding Agents* in parallel, assigning each a distinct draft and all referenced resources. Upon completion, the *Coordinator* aggregates the results and publishes successful solutions back to the community, advancing the environment state for the next iteration.

**Analyzer** The *Analyzer* is responsible for distilling raw community artifacts into structured, actionable intelligence. It receives the sampled kernels and discussions from the *Coordinator* and performs a deep analysis across four key dimensions: novelty, feasibility, effectiveness, and efficiency. For each artifact, it generates a 0-10 score on these metrics, accompanied by qualitative explanations of successful patterns, emerging trends, or potential pitfalls. The output is a set of structured analytical reports $\mathcal{R}_t$, which serve as the primary input for the *Idea Proposer*.

**Idea Proposer** The *Idea Proposer* functions as **CoMind**'s creative engine, tasked with generating novel solution concepts. It uses the analytical reports $\mathcal{R}_t$ from the *Analyzer* and its own persistent memory of historical ideas $\mathcal{I}^*_t$ to ensure that new concepts are both innovative and informed by past results. The ideation process follows three phases: (1) **Brainstorming:** Generating a wide array of diverse ideas, prioritizing creativity and exploration. (2) **Filtering:** Ranking these ideas based on feasibility, potential for improvement, and alignment with the analytical reports. Only the most promising subset of ideas $\mathcal{I}_t$ is selected. (3) **Memory Integration:** Updating its knowledge base with the newly generated ideas ($\mathcal{I}^*_{t+1} = \mathcal{I}^*_t \cup \mathcal{I}_t$), allowing for increasingly sophisticated strategies over time. The final output, a filtered set of high-potential ideas $\mathcal{I}_t$, is sent back to the *Coordinator* to be developed into full solution drafts.

**Coding Agent** The *Coding Agent* is the implementation workhorse, responsible for converting the abstract solution drafts from the *Coordinator* into executable code. Following an iterative, ReAct-style approach, it conducts trial-and-error experiments using the training and validation data provided by the *Evaluator*. To maximize efficiency, the agent maintains a persistent Jupyter Notebook session to eliminate data reloading overhead and employs a monitor LLM to track execution and terminate failed runs immediately. This iterative process of coding, debugging, and optimization continues until a viable solution is produced or a time budget is exhausted.

**Evaluator** The *Evaluator* ensures objective, standardized, and reproducible assessment across all experiments, mirroring official Kaggle protocols. It first partitions the public dataset $D$ into a training set $D^*$ and a validation set with inputs $V_x$ and ground-truth labels $V_y$. Crucially, only $D^*$ and $V_x$ are accessible to the *Coding Agents*, preserving the integrity of the validation process. When a *Coding Agent* submits predictions $V_{\hat{y}}$, the *Evaluator* computes the performance score using the official competition metric $\varphi(V_{\hat{y}}, V_y)$. It maintains a global leaderboard of all experimental runs, enabling **CoMind** to reliably track progress and make informed decisions about which solutions to prioritize and publish.

## 5 BENCHMARK EVALUATION

### 5.1 SETUP

**Task Selection.** Based on MLE-Live evaluation framework, we evaluate our agent on 75 Kaggle competitions on MLE-Bench. Using the MLE-Live framework, CoMind has access to shared discussions and public kernels published on the competition websites before the competition deadline. Since the MLE-bench test set may be constructed from Kaggle's official public training set, and publicly available datasets or model checkpoints may have been trained on this portion of the data,

Table 1: Any Medal (%) scores on 75 MLE-Bench competitions. CoMind achieves state-of-the-art results across difficulty levels. Best results in each column are bolded. Baseline numbers are taken from the official MLE-Bench leaderboard.

| Agent | Low (%) | Medium (%) | High (%) | All (%) |
|---|---|---|---|---|
| **CoMind** o4-mini | **59.09** | 23.68 | **33.33** | **36.00** |
| Neo multi-agent | 48.48 | **29.82** | 24.44 | 34.22 |
| R&D-Agent o3 + GPT-4.1 | 51.52 | 19.30 | 26.67 | 30.22 |
| ML-Master deepseek-r1 | 48.50 | 20.20 | 24.40 | 29.30 |
| R&D-Agent o1-preview | 48.18 | 8.95 | 18.67 | 22.40 |
| AIDE o1-preview | 34.30 | 8.80 | 10.00 | 16.90 |
| AIDE gpt-4o | 19.00 | 3.20 | 5.60 | 8.60 |
| AIDE claude-3-5-sonnet | 19.40 | 2.60 | 2.30 | 7.50 |
| OpenHands gpt-4o | 11.50 | 2.20 | 1.90 | 5.10 |
| AIDE llama-3.1-405b-instruct | 8.30 | 1.20 | 0.00 | 3.10 |
| MLAB gpt-4o | 4.20 | 0.00 | 0.00 | 1.30 |

we restricted CoMind's access to public datasets to minimize potential data contamination. It can only view code published by other contestants.

To validate CoMind under realistic conditions, we further evaluate CoMind on eight ongoing Kaggle competitions. These competitions span diverse domains, including tabular learning, text regression, image classification and video recognition. Rather than approximating the official scoring locally, we directly submit CoMind's generated `submission.csv` files to the Kaggle platform, so that all reported ranks reflect genuine, live leaderboard positions.

**Implementation Details.** CoMind employs `o4-mini-2025-04-16` (OpenAI, 2025) as its backend LLM. We limit the hardware constraint of each run to 32 vCPUs and a single A6000 GPU. Each competition is evaluated in separate containers with a maximum of 24 hours to produce the final submission file. Every single code execution session is limited to 5 hour. Each Coder is limited to a maximum of 30 steps. The number of parallel agents is set to 4.

During code generation, agents are provided with the test set inputs (without labels) and prompted to generate a `submission.csv` file. The submission is then evaluated by a grader that compares the predicted labels with the ground truth. Following the setting of MLE-Bench, to avoid potential overfitting, test set labels and the competition leaderboard are strictly withheld from the agent's accessible environment. Instead, each agent must rely solely on a self-constructed "runtime test set", a held-out split from the original training data, for code evaluation and performance estimation.

**Metrics.** Following the evaluation metrics in MLE-Bench, we measure the performance of Co-Mind by **Any Medal**, the percentage of competitions where the agent earns a gold, silver, or bronze medal.

**Baselines.** We compare CoMind against the MLE-Bench leaderboard[1] including open-sourced systems like **R&D-Agent** (Yang et al., 2025), a dual-agent framework (Researcher/Developer) that explores multiple solution branches and merges promising ideas into improved pipelines; **ML-Master** (Liu et al., 2025), which integrates exploration and reasoning via a selectively scoped memory that aggregates insights from parallel trajectories; **AIDE** (Jiang et al., 2025), a purpose-built tree-search scaffold that iteratively drafts, debugs, and benchmarks code for Kaggle-style tasks; **OpenHands** (Wang et al., 2025), a general-purpose CodeAct-based scaffold that executes code and calls tools in a sandboxed environment; **MLAB** (Huang et al., 2024), referring to the ResearchAgent scaffold from MLAgentBench, a general tool-calling/plan–act baseline; and **Neo** (https://heyneo.so/), a close-sourced multi-agent system for autonomous ML engineering.

---

[1]https://github.com/openai/mle-bench

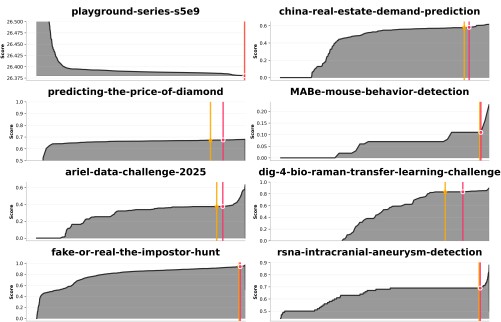

| Competition | Rank | Teams | Top % |
|---|---|---|---|
| Playground S5E9 | 4 | 1966 | 0.2% |
| China Real Estate | 43 | 437 | 9.8% |
| Diamond Price | 8 | 67 | 11.9% |
| MABe Behavior | 3 | 51 | 5.9% |
| ARIEL 2025 | 90 | 827 | 10.9% |
| DIG4BIO Raman TL | 22 | 167 | 13.2% |
| Impostor Hunt | 26 | 1037 | 2.5% |
| RSNA Aneurysm | 35 | 788 | 4.4% |

Figure 3: **Left:** Score distributions across participants in eight ongoing Kaggle competitions. Each curve shows the relationship between leaderboard rank (x-axis, inverted) and competition score (y-axis). Vertical lines indicate CoMind's position (red) and public best performance (yellow). **Right:** Results on eight ongoing Kaggle competitions. Reported are leaderboard rank, total teams, and percentile rank (Top %, where lower means better standing).

## 5.2 RESULTS

Table 1 compares CoMind with baseline methods on 75 MLE-Bench competitions. CoMind achieves state-of-the-art performance with an *Any Medal* rate of 36.00%, significantly outperforming open-source competitors such as R&D-Agent (submitted on 2025-08-15) and surpassing the closed-source multi-agent system Neo. Appendix J provides a detailed case study on denoising-dirty-documents.

On the eight evaluated ongoing competitions, CoMind ranked top 7.35% on average and improved the best public kernel on 5 competitions. Details including authentic scores and win rates per task are provided in Figure 3. These authentic results demonstrate CoMind's capability to tackle a variety of problem domains and achieve competitive performance in live, evolving ML workflows.

## 6 ABLATION STUDY

### 6.1 SETUP

**Task Selection.** To evaluate the impact of introducing public resources, we conducted an ablation study on 20 competitions from MLE-Bench-Lite based on MLE-Live. These tasks span across various categories, including image classification/generation, text classification/generation, image regression, audio classification, and tabular analysis.

**Baselines.** We compared CoMind against the following baselines. For consistency, all baselines use the same backend model as CoMind:

- **AIDE+Code.** To enable the use of publicly available code (e.g., Kaggle kernels), we extend AIDE with access to one public kernel per draft node, which is selected by highest community votes. AIDE+Code augments the prompt with both the task description and the selected kernel alongside the tree summarization.
- **AIDE+RAG.** We further equip AIDE with a retrieval-augmented generation (RAG) mechanism. Before generating code, the agent retrieves the titles of the top 10 voted discussions and kernels. The LLM selects the most relevant ones, receives a summarization, and then proposes its plan and implementation. For debugging or refinement, it can optionally re-query documents. Retrieval is based on cosine similarity between query and candidate document embeddings, using Multilingual E5 Text Embeddings (Wang et al., 2024).
- **CoMind w/o $\mathcal{R}$.** $\mathcal{R}$ denotes all public resources. In this variant, CoMind operates without access to any external community resources. It starts with an empty community and relies solely on its own generation history to propose candidate ideas and assemble solution drafts.

**Metrics.** Following the evaluation metrics in prior research (Chan et al., 2025), the relative capability of generating high-quality solution compared with human is measured by:

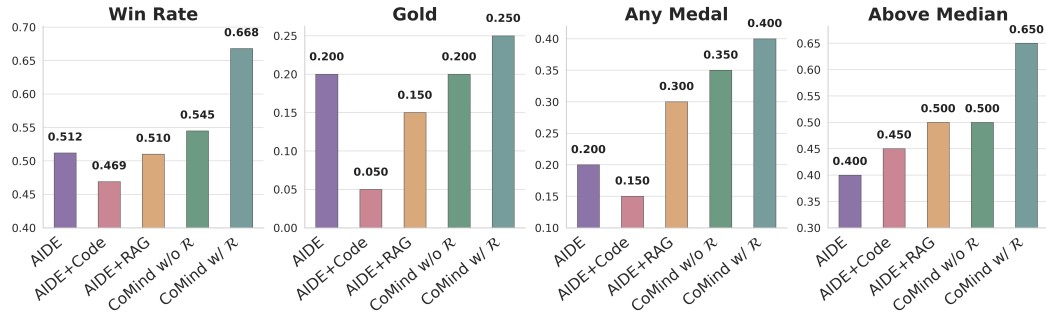

Figure 4: **Performance of CoMind and other baselines on 20 competitions from MLE-Bench-Lite.** *Valid Submission* is the ratio of submissions meeting format requirements and validation criteria. *Win Rate* is the percentage of human competitors outperformed by the agent. *Any Medal*, is the proportion of competitions where the agent earned Gold, Silver or Bronze medals. *Above Median* is the fraction of competitions where the agent's score strictly exceeded the median human competitor.

Table 2: **Average win rate of CoMind and other baselines across task categories on 20 competitions from MLE-Bench-Lite.** *# of Tasks* refers to the number of competitions in the corresponding category. CoMind consistently outperforms baselines across most domains.

| Category | # of Tasks | CoMind | AIDE+Code | AIDE+RAG | AIDE |
|---|---|---|---|---|---|
| Image Classification | 8 | **0.597** | 0.459 | 0.434 | 0.525 |
| Text Classification | 3 | **0.740** | 0.157 | 0.338 | 0.61 |
| Audio Classification | 1 | **0.901** | 0.272 | 0.259 | 0.271 |
| Seq2Seq | 2 | 0.408 | 0.503 | **0.550** | 0.228 |
| Tabular | 4 | 0.664 | 0.673 | **0.688** | 0.483 |
| Image To Image | 1 | **0.988** | 0.932 | 0.617 | 0.568 |
| Image Regression | 1 | **0.992** | 0.342 | **0.992** | **0.992** |
| All | 20 | **0.668** | 0.469 | 0.510 | 0.512 |

- **Above Median**: Indicates whether the submission outperforms at least 50% of competitors on the leaderboard.
- **Win Rate**: The percentage of competitors whose final scores are lower than the agent's score. If the agent fails to produce a valid submission, the Win Rate is 0.
- **Medals**: Medals are assigned based on the agent's score relative to Kaggle leaderboard thresholds for gold, silver, and bronze medals.
- **Any Medal**: The percentage of competitions in which the agent earns any medal.

**Implementation Setup.** All agents use `o4-mini-2025-04-16` as their backend. Based on the settings of our main experiment, the hardware constraint is further limited to 4 vCPUs and 5 hours per competition. Each execution session is limited to 1 hour. Access to public datasets are restricted. In accordance with baselines, CoMind has access to 10 top-voted discusions and kernels.

## 6.2 RESULTS

Figure 4 shows the results. Our key findings are as follows: (i) CoMind consistently outperforms all baselines across every metric. (ii) Among the AIDE variants, AIDE+RAG outperforms AIDE+Code, and both surpass the original AIDE on most metrics, demonstrating the benefits of integrating community knowledge. CoMind further exceeds these approaches, highlighting the effectiveness of its deeper and more strategic community-aware exploration. (iii) Removing CoMind's resource access causes a significant drop in valid submission rates and other metrics, showing that strategic access to public resources helps CoMind balance extending established methods for reliability with exploring novel approaches.

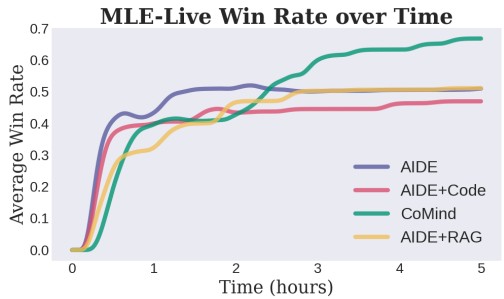
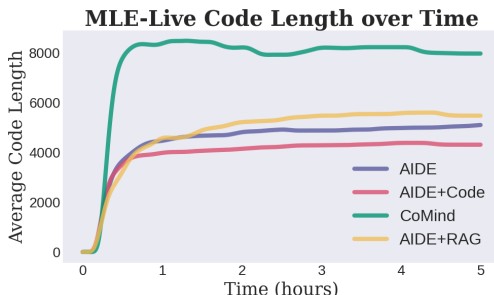

Figure 5: Win rate over time. CoMind sustains improvement while baselines plateau.

Figure 6: Code complexity over time. CoMind generates longer, richer solutions than baselines.

## 7 ANALYTICAL EXPERIMENTS

For analytical experiments, we adopt the same setup as the ablation study and evaluate model performance across multiple dimensions, including task categories, win rate over time, and code complexity.

**Task Categories**    Table 2 reports the average ranks across seven task categories. CoMind outperforms all baselines in Image Classification, Text Classification, Audio Classification, and Image-to-Image tasks, highlighting its strong adaptability. We manually inspect the tasks where CoMind underperformed and find that the issues are often related to the use of large models or datasets. For example, in Seq2Seq tasks, CoMind explores complex fine-tuning strategies for large language models which often fail to complete within the one-hour runtime constraint.

**Win Rate Over Time**    Figure 5 shows the evolution of average win rate over time. AIDE quickly produces concise, functional solutions, leading to a rapid rise in performance during the first hour. In contrast, CoMind spends more time on debugging and exploration early on, resulting in a slower initial improvement. However, after the first two hours, AIDE's performance plateaus, while CoMind continues to improve through iterative refinement and deeper exploration, ultimately surpassing AIDE and achieving higher-quality solutions.

**Code Complexity**    Regarding code complexity, Figure 6 illustrates the average code length during the entire competition. CoMind consistently generates significantly longer and more complex code, while other baselines begin with simpler implementations and introduce only incremental modifications. Appendix B offers a comparative analysis across code complexity metrics and task categories. Notably, CoMind's solutions for Image Regression and Audio Classification are nearly twice as long as those of other baselines. Additionally, solutions from CoMind are, on average, 55.4% longer than those produced by AIDE.

## 8 CONCLUSION

We introduced MLE-Live, the first framework to evaluate ML agents in community-driven settings, simulating the collaborative dynamics that are essential to real-world progress in Kaggle competitions and beyond. Building upon this benchmark, we proposed CoMind, a community-augmented LLM agent that iteratively selects and synthesizes ideas, implements solutions, and shares reports within a simulated ecosystem. Our results demonstrate that CoMind not only achieves state-of-the-art performance on retrospective MLE-Bench tasks but also attains medal-level standings in live Kaggle competitions.

**Limitations and Future Work.**    While our current experiments focus on Kaggle-style ML tasks, the MLE-Live framework can be extended to broader domains, such as scientific discovery, open-ended coding, or robotics, enabling research agents to contribute meaningfully across diverse fields.

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

# A   USE OF LLMS

We employ large language models (LLMs) exclusively for the purpose of assisting in the drafting and refinement of our manuscripts, with the objective of enhancing clarity and coherence.

# B   ADDITIONAL ANALYSIS ON CODE COMPLEXITY

In this section, we provide a comprehensive analysis of the generated code using a broad set of software complexity and quality metrics, beyond mere line counts. Specifically, we report the following indicators: **Cyclomatic Complexity (CC)**, **Pylint score**, **Halstead Metrics**: Volume, Difficulty, Effort, **Source Lines of Code (SLOC)**, **Number of Comment Lines** and **Code Length**. We prioritized these over human annotation to ensure reproducibility and avoid subjective bias.

Table 3: **Code complexity and quality metrics (Cyclomatic Complexity, Pylint score, Halstead metrics, SLOC, etc.) across task categories.** CoMind produces more complex solutions compared to baselines.

| Category | Metric | CoMind | AIDE | AIDE+RAG | AIDE+Code |
|---|---|---|---|---|---|
| Image Classification | CC | 1.68 | 1.59 | 1.93 | 1.29 |
| | Pylint Score | 7.43 | 9.06 | 8.90 | 8.92 |
| | Volume | 330.88 | 143.26 | 84.20 | 175.88 |
| | Difficulty | 4.95 | 2.90 | 2.32 | 2.59 |
| | Effort | 1960.22 | 507.06 | 286.31 | 725.59 |
| | SLOC | 198.25 | 133.50 | 120.88 | 115.71 |
| | Comment Lines | 15.62 | 12.88 | 13.75 | 14.43 |
| | Code Length | 7638.40 | 4624.30 | 4701.30 | 5192.10 |
| Text Classification | CC | 3.58 | 4.28 | 2.00 | 0.00 |
| | Pylint Score | 8.82 | 9.09 | 8.89 | 9.26 |
| | Volume | 286.38 | 384.07 | 47.68 | 29.25 |
| | Difficulty | 3.76 | 3.94 | 1.25 | 1.31 |
| | Effort | 1183.11 | 2332.22 | 61.56 | 35.16 |
| | SLOC | 181.67 | 133.00 | 141.00 | 69.50 |
| | Comment Lines | 14.67 | 15.33 | 14.00 | 13.50 |
| | Code Length | 6974.70 | 3094.50 | 5920.50 | 5629.30 |
| Audio Classification | CC | 2.00 | 0.00 | 0.00 | 0.00 |
| | Pylint Score | 7.92 | 9.11 | 9.49 | 8.86 |
| | Volume | 718.63 | 244.20 | 115.95 | 227.48 |
| | Difficulty | 7.39 | 6.46 | 3.19 | 6.38 |
| | Effort | 5308.07 | 1577.11 | 369.58 | 1451.30 |
| | SLOC | 256.00 | 82.00 | 92.00 | 72.00 |
| | Comment Lines | 20.00 | 11.00 | 16.00 | 16.00 |
| | Code Length | 9449.00 | 3508.00 | 4151.00 | 3352.00 |
| Seq2Seq | CC | 4.38 | 2.25 | 22.33 | 15.75 |
| | Pylint Score | 8.58 | 9.04 | 9.14 | 8.51 |
| | Volume | 492.55 | 52.33 | 390.46 | 324.00 |
| | Difficulty | 3.87 | 2.14 | 5.26 | 3.68 |
| | Effort | 1935.02 | 140.58 | 2083.84 | 1686.74 |
| | SLOC | 184.50 | 63.50 | 222.50 | 147.50 |
| | Comment Lines | 22.50 | 13.00 | 23.00 | 19.50 |
| | Code Length | 6925.50 | 5649.50 | 8357.50 | 2728.50 |
| Tabular | CC | 2.78 | 1.62 | 2.38 | 0.25 |
| | Pylint Score | 8.65 | 8.96 | 8.87 | 9.31 |
| | Volume | 1264.61 | 856.12 | 815.29 | 435.46 |
| | Difficulty | 7.37 | 4.83 | 6.05 | 3.69 |
| | Effort | 10 808.93 | 6163.62 | 5564.22 | 2001.06 |
| | SLOC | 218.75 | 139.75 | 147.50 | 93.50 |
| | Comment Lines | 18.25 | 14.75 | 15.25 | 10.50 |
| | Code Length | 8570.00 | 3534.00 | 6064.00 | 5759.80 |

| Category | Metric | CoMind | AIDE | AIDE+RAG | AIDE+Code |
|---|---|---|---|---|---|
| | CC | 1.72 | 2.00 | 3.00 | 1.88 |
| | Pylint Score | 8.43 | 6.25 | 6.64 | 7.74 |
| | Volume | 1298.11 | 1481.62 | 414.59 | 431.08 |
| | Difficulty | 9.68 | 6.73 | 3.94 | 3.79 |
| Image to Image | Effort | 12 565.66 | 9967.24 | 1633.22 | 1631.93 |
| | SLOC | 228.00 | 175.00 | 121.00 | 128.00 |
| | Comment Lines | 26.00 | 8.00 | 23.00 | 13.00 |
| | Code Length | 8800.00 | 5231.00 | 4815.00 | 6671.00 |
| | CC | 1.68 | 2.00 | 2.40 | 2.00 |
| | Pylint Score | 8.62 | 8.75 | 8.80 | 8.89 |
| | Volume | 1310.92 | 241.08 | 70.32 | 72.00 |
| | Difficulty | 8.75 | 3.88 | 2.18 | 2.73 |
| Image Regression | Effort | 11 466.58 | 934.17 | 153.43 | 196.36 |
| | SLOC | 267.00 | 145.00 | 116.00 | 133.00 |
| | Comment Lines | 36.00 | 15.00 | 12.00 | 12.00 |
| | Code Length | 10 991.00 | 4841.00 | 4655.00 | 5614.00 |

## C  TOKEN USAGE ON MLE-BENCH COMPETITIONS

| Agent | Uncached Tokens | Cached Tokens | Completion Tokens | Cost |
|---|---|---|---|---|
| CoMind | 18.23 M ± 14.62 M | 13.05 M ± 11.43 M | 1.96 M ± 1.04 M | $32.25 ± 19.43 |

Table 4: **Average token usage (mean ± standard deviation) and monetary cost for CoMind across the 75 Kaggle competitions in MLE-Bench.** The cost is calculated based on the API prices of o4-mini published by OpenAI.

## D  LOCAL EVALUATION DYNAMICS OF COMIND ON ONGOING COMPETITIONS

Figure 7 plot the agent's locally estimated validation metrics over its execution time, as computed by the Evaluator module using held-out validation splits constructed from the public training data. These trajectories capture how CoMind iteratively explores, implements, debugs, and refines solution pipelines through its multi-agent workflow. It illustrates several consistent patterns: (i) rapid early improvements as baseline solutions are assembled; (ii) mid-phase refinements driven by Analyzer-guided idea selection and iterative debugging; and (iii) late-stage stabilization once the agent converges to high-performing configurations.

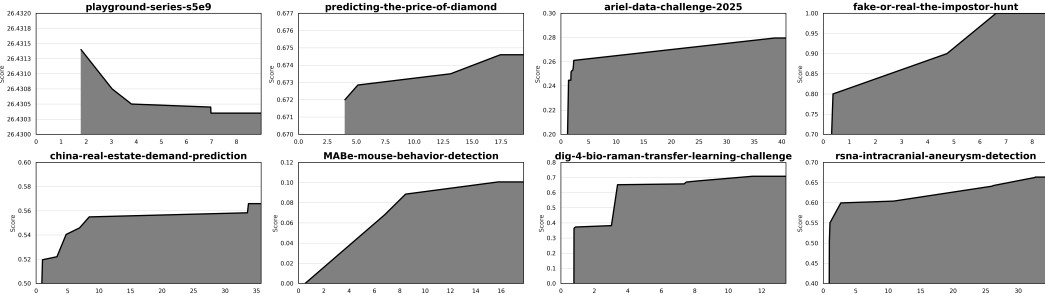

Figure 7: Evolution of CoMind's Locally Estimated Competition Metrics Over Runtime Across Eight Ongoing Kaggle Tasks.

# E  CATEGORIES AND DIFFICULTIES IN MLE-BENCH

MLE-Bench (Chan et al., 2025) curates 75 ML engineering-related competitions from Kaggle, creating a diverse set of challenging tasks that test real-world ML engineering skills. These competitions span 15 diverse problem categories. Each competition has an associated description, dataset, and grading code. MLE-Bench categorizes competitions based on human evaluation results: Low (29%) if an experienced ML engineer can produce a sensible solution in 2 hours (excluding training time), Medium (51%) for 2–10 hours, and High (20%) for more than 10 hours.

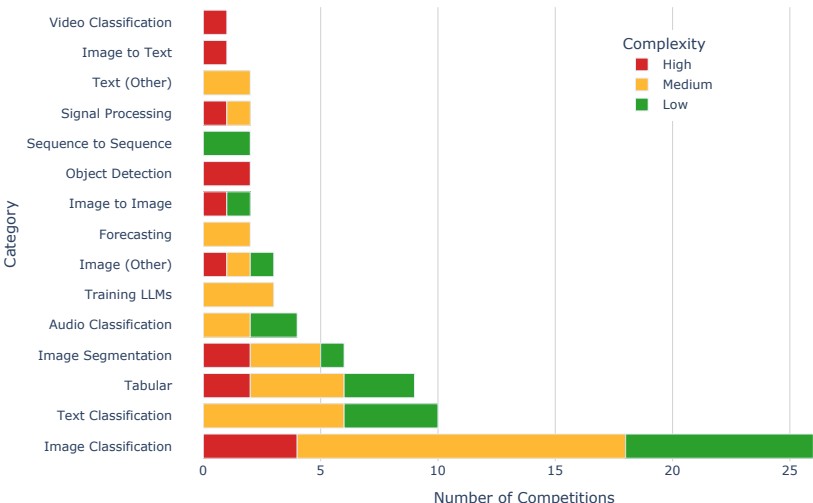

Figure 8: (From MLE-Bench) **Competitions in MLE-Bench spans 15 categories.** MLE-Bench categorizes competition difficulties in Low, Medium and High.

# F  COMIND ON COBENCH

To directly assess CoMind's extensibility beyond Kaggle-style environments, we conducted an additional experiment on CoBench (Sun et al., 2025), a dataset of 36 combinatorial optimization problems that is structurally different from Kaggle competitions and does not involve notebooks, discussions, or leaderboard dynamics. Using gpt-5-mini as a shared backbone, we compared CoMind (without access to public resources) against AIDE (Jiang et al., 2025), FunSearch (Romera-Paredes et al., 2024), and Greedy Refinement (iteratively prompting the LLM to refine the current best solution). Following the settings of CoBench, each agent operates through 64 research steps, receiving feedbacks on a test set at each step and has a 10-second execution time limitation. Across 9 representative CoBench tasks, CoMind's normalized score achieves competitive performance and, on average, outperforms the baselines, as shown in Table 5.

# G  MLE-LIVE ON MLE-BENCH COMPETITIONS

We collected all public kernels and discussions posted before each competition's deadline, preserving the temporal information to simulate realistic research environments. Table 6 presents detailed statistics for each competition in MLE-Live. The dataset encompasses 15,733 kernels and 12,951 discussions across competitions of varying difficulty levels (Low, Medium, High).

Figure 9 shows the distribution of kernel votes across all competitions. The distribution is heavily skewed, with over half of the kernels receiving fewer than 10 votes. This long-tail pattern reflects real-world community dynamics where a small fraction of high-quality contributions attracts substantial attention, while most submissions receive minimal engagement.

| Task | CoMind | AIDE | FunSearch | Greedy Refinement |
|---|---|---|---|---|
| Crew scheduling | **0.915** | 0.448 | 0.546 | 0.602 |
| Graph colouring | 0.879 | 0.850 | 0.893 | **0.968** |
| Constrained guillotine cutting | 0.975 | 0.911 | **0.993** | 0.989 |
| MIS | 0.909 | **0.932** | 0.860 | 0.874 |
| Aircraft landing | **0.865** | 0.863 | 0.760 | 0.378 |
| Bin packing (1D) | 0.903 | 0.925 | **0.975** | 0.821 |
| Euclidean Steiner problem | 0.690 | 0.636 | 0.701 | **0.760** |
| Set covering | **0.922** | 0.887 | 0.918 | 0.916 |
| TSP | **0.923** | 0.606 | 0.860 | 0.832 |
| **Avg** | **0.887** | 0.784 | 0.834 | 0.793 |

Table 5: **Evaluation on 9 representative task of CoBench.**

Table 6: **Statistics of MLE-Live on MLE-Bench competitions.** *Kern.* is the number of public kernels. *Disc.* is the number of discussions. *Dsets* is the number of external datasets referenced by public kernels, *Lines* refers to the average number of lines of all kernels, *Comms* is the average number of comments in all discussions.

| Competition | Kern. | Disc. | Dsets | Lines | Comms |
|---|---|---|---|---|---|
| **Low** | | | | | |
| aerial-cactus-identification | 275 | 27 | 12 | 177.40 | 3.52 |
| aptos2019-blindness-detection | 186 | 503 | 70 | 357.18 | 8.33 |
| denoising-dirty-documents | 59 | 19 | 1 | 70.36 | 4.53 |
| detecting-insults-in-social-commentary | 0 | 27 | 0 | – | 4.26 |
| dog-breed-identification | 64 | 33 | 9 | 161.75 | 5.15 |
| dogs-vs-cats-redux-kernels-edition | 0 | 31 | 0 | – | 4.68 |
| histopathologic-cancer-detection | 118 | 51 | 12 | 228.19 | 9.53 |
| jigsaw-toxic-comment-classification-challenge | 128 | 394 | 56 | 80.88 | 7.19 |
| leaf-classification | 803 | 23 | 1 | 143.37 | 1.61 |
| mlsp-2013-birds | 0 | 35 | 0 | – | 4.94 |
| new-york-city-taxi-fare-prediction | 143 | 53 | 6 | 256.69 | 5.75 |
| nomad2018-predict-transparent-conductors | 53 | 40 | 9 | 57.14 | 5.30 |
| plant-pathology-2020-fgvc7 | 207 | 91 | 27 | 281.84 | 5.14 |
| random-acts-of-pizza | 8 | 19 | 1 | 35.25 | 3.74 |
| ranzcr-clip-catheter-line-classification | 323 | 289 | 156 | 326.17 | 6.99 |
| siim-isic-melanoma-classification | 276 | 707 | 135 | 345.59 | 10.87 |
| spooky-author-identification | 153 | 68 | 13 | 103.45 | 3.75 |
| tabular-playground-series-dec-2021 | 427 | 134 | 53 | 233.22 | 6.84 |
| tabular-playground-series-may-2022 | 247 | 64 | 23 | 273.53 | 5.02 |
| text-normalization-challenge-english-language | 57 | 51 | 3 | 153.93 | 4.45 |
| text-normalization-challenge-russian-language | 12 | 20 | 2 | 37.67 | 2.85 |
| the-icml-2013-whale-challenge-right-whale-redux | 0 | 8 | 0 | – | 1.62 |
| **Medium** | | | | | |
| AI4Code | 159 | 178 | 93 | 395.78 | 5.76 |
| alaska2-image-steganalysis | 109 | 154 | 31 | 317.38 | 6.35 |
| billion-word-imputation | 0 | 23 | 0 | – | 3.43 |
| cassava-leaf-disease-classification | 411 | 724 | 209 | 358.37 | 6.77 |
| cdiscount-image-classification-challenge | 91 | 109 | 3 | 57.10 | 9.25 |
| chaii-hindi-and-tamil-question-answering | 269 | 147 | 137 | 341.48 | 6.74 |
| champs-scalar-coupling | 290 | 340 | 49 | 307.36 | 8.54 |
| facebook-recruiting-iii-keyword-extraction | 0 | 72 | 0 | – | 5.32 |
| freesound-audio-tagging-2019 | 109 | 128 | 18 | 349.46 | 7.23 |
| google-quest-challenge | 225 | 258 | 128 | 442.04 | 9.04 |
| h-and-m-personalized-fashion-recommendations | 419 | 223 | 63 | 240.76 | 6.30 |

| Competition | Kern. | Disc. | Dsets | Lines | Comms |
|---|---|---|---|---|---|
| herbarium-2020-fgvc7 | 19 | 15 | 3 | 205.83 | 3.67 |
| herbarium-2021-fgvc8 | 21 | 26 | 17 | 266.86 | 2.46 |
| herbarium-2022-fgvc9 | 36 | 37 | 6 | 274.52 | 3.84 |
| hotel-id-2021-fgvc8 | 30 | 30 | 15 | 196.00 | 2.07 |
| hubmap-kidney-segmentation | 321 | 340 | 178 | 325.47 | 6.70 |
| icecube-neutrinos-in-deep-ice | 187 | 156 | 38 | 319.70 | 5.58 |
| imet-2020-fgvc7 | 32 | 21 | 13 | 298.37 | 3.67 |
| inaturalist-2019-fgvc6 | 12 | 11 | 3 | 244.17 | 4.27 |
| iwildcam-2020-fgvc7 | 21 | 22 | 5 | 256.86 | 3.55 |
| jigsaw-unintended-bias-in-toxicity-classification | 410 | 413 | 118 | 301.86 | 8.16 |
| kuzushiji-recognition | 42 | 55 | 4 | 226.57 | 5.00 |
| learning-agency-lab-automated-essay-scoring-2 | 477 | 226 | 301 | 349.24 | 7.78 |
| lmsys-chatbot-arena | 305 | 260 | 123 | 299.33 | 8.51 |
| multi-modal-gesture-recognition | 0 | 39 | 0 | – | 3.36 |
| osic-pulmonary-fibrosis-progression | 513 | 499 | 80 | 386.23 | 5.92 |
| petfinder-pawpularity-score | 397 | 461 | 153 | 240.50 | 5.86 |
| plant-pathology-2021-fgvc8 | 325 | 112 | 135 | 273.29 | 4.20 |
| seti-breakthrough-listen | 230 | 222 | 72 | 333.81 | 7.82 |
| statoil-iceberg-classifier-challenge | 160 | 178 | 15 | 80.98 | 6.09 |
| tensorflow-speech-recognition-challenge | 47 | 139 | 4 | 55.94 | 6.83 |
| tensorflow2-question-answering | 102 | 193 | 52 | 420.29 | 6.71 |
| tgs-salt-identification-challenge | 213 | 336 | 24 | 329.40 | 12.16 |
| tweet-sentiment-extraction | 536 | 387 | 117 | 324.96 | 9.86 |
| us-patent-phrase-to-phrase-matching | 456 | 233 | 352 | 284.25 | 7.15 |
| uw-madison-gi-tract-image-segmentation | 233 | 230 | 124 | 445.30 | 6.32 |
| ventilator-pressure-prediction | 407 | 268 | 70 | 246.07 | 9.05 |
| whale-categorization-playground | 32 | 22 | 4 | 61.93 | 5.00 |
| **High** | | | | | |
| 3d-object-detection-for-autonomous-vehicles | 39 | 116 | 10 | 565.09 | 6.19 |
| bms-molecular-translation | 167 | 218 | 57 | 463.33 | 10.39 |
| google-research-identify-contrails-reduce-global-warming | 143 | 136 | 115 | 322.43 | 6.24 |
| hms-harmful-brain-activity-classification | 631 | 390 | 258 | 573.88 | 6.64 |
| iwildcam-2019-fgvc6 | 37 | 26 | 9 | 196.06 | 3.77 |
| nfl-player-contact-detection | 94 | 81 | 32 | 342.78 | 4.17 |
| predict-volcanic-eruptions-ingv-oe | 102 | 55 | 21 | 271.21 | 3.55 |
| rsna-2022-cervical-spine-fracture-detection | 209 | 152 | 106 | 302.68 | 4.34 |
| rsna-breast-cancer-detection | 538 | 433 | 291 | 342.78 | 7.11 |
| rsna-miccai-brain-tumor-radiogenomic-classification | 414 | 334 | 131 | 321.36 | 5.72 |
| siim-covid19-detection | 586 | 419 | 382 | 337.11 | 5.59 |
| smartphone-decimeter-2022 | 49 | 56 | 16 | 244.82 | 3.79 |
| stanford-covid-vaccine | 201 | 194 | 29 | 342.38 | 8.55 |
| vesuvius-challenge-ink-detection | 190 | 177 | 72 | 387.63 | 6.19 |
| vinbigdata-chest-xray-abnormalities-detection | 317 | 187 | 115 | 379.55 | 6.70 |

# H    ERROR ANALYSIS

In this section, we analyze why CoMind failed to surpass the strongest public kernel in the `fake-or-real-the-impostor-hunt` Kaggle competition. The task involves identifying the fake text within each pair of samples. Although the dataset originates from The Messenger journal, both the "real" and "fake" texts have been heavily modified by LLMs, making the distinction more subtle and challenging. Below, we summarize the main obstacles encountered during CoMind's execution:

**Noised external resources.**    Public voting did not reliably indicate kernel quality; many highly upvoted kernels were merely ensembles or reused outputs from stronger solutions, offering little actionable insight. With a large volume of heterogeneous public contributions, identifying genuinely informative resources remained difficult.

**Sparse evaluation signal.**    CoMind depends on the Evaluator's feedback to guide iteration, but the task's extremely small dataset allowed validation on only 10 examples. This produced highly unstable feedback and limited the system's ability to differentiate between small performance variations. As shown in Figure 7,

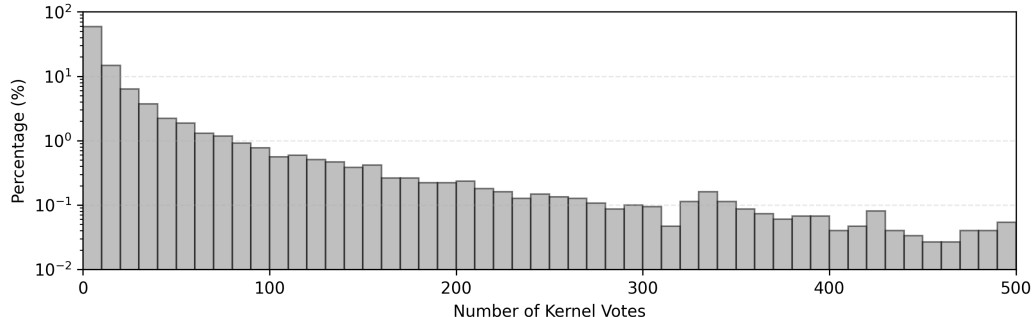

Figure 9: **Distribution of kernel votes across all competitions in MLE-Bench.** Over half of the kernels received fewer than 10 votes, demonstrating the long-tail nature of community contributions where most submissions receive minimal engagement while a small fraction attracts substantial attention.

Evaluator accuracy saturated at 100% within the first 7 hours, leaving no meaningful gradient for further improvement.

**Insufficient ablation analysis.** While CoMind explored multiple strategies, its assessment of individual module effectiveness was inconsistent. For example, early iterations attempted to fine-tune local LLMs via LoRA and train binary classifiers on their outputs, but due to poor base model selection and insufficient hyperparameter exploration, these approaches underperformed simple classical signals such as TF-IDF cosine similarity and token-level similarity.

# I   PROMPTS AND RESPONSES FOR COMIND

This section provides some examples of prompts and responses in CoMind, including **Coordinator**, **Analyzer**, **Idea Proposer**, **Coding Agent** and **Evaluator**.

## I.1   COORDINATOR

---

**Prompt for Solution Draft Synthesis**

**Introduction**   You are an expert machine learning researcher preparing for the Kaggle competition described below.
**Task Description**   {*description of the specified task*}
**Ideas**   {*entries in the idea pool*}
**Reports**   {*entries in the report pool*}
**Public Pipelines**   {*all public pipelines extracted before*}
**Goals**

1. Carefully read the reports provided above.

2. Based on the ideas and reports, propose {*num_pipes*} **promising self-contained pipelines** that are likely to perform well.

3. The Public pipelines section contains top-ranked public pipelines during the competition. Use them as reference to polish your pipelines.

4. Each pipeline should not overlap with others. Your proposed pipelines should include **one baseline pipeline that uses well-known methods but is robust and relatively easy to implement**. You should reinforce public pipelines and previous pipelines based on their reports (if provided).

5. Ensure that each pipeline can be trained within 2 hours on a single A6000 with 48GB memory.

6. Read the **submission format** requirements in the task description carefully. The format requirement is possible to be different from the training dataset. **THIS IS EXTREMELY IMPORTANT**. Mention in the pipeline descriptions and be sure to include the code that handles the input and output.

7. DO NOT USE tensorflow, use pytorch instead

---

**Response Template for Solution Draft Synthesis**

**Submit Pipelines** Descriptions and codes of pipelines, separated each pipeline by ===SEPARA-TOR=== mark. For each pipeline, attach code that captures its essential. **You must include the code in public pipelines that handles input and output, and if there are parts of the public pipelines that are similar to the current pipeline, you should include them as well.**

---

## I.2   ANALYZER

---

**Prompt for Strategy Distilation**

**Introduction**    You are an expert machine learning researcher preparing for the Kaggle competition described below.

**Task Description**    {*description of the specified task*}

**Goals**    These are top-ranked public scripts during the competition. Your job is to:

1. Carefully read the following scripts.

2. For each script, if it's self-contained, i.e., including model architecture (if there's a model), training strategies, evaluation, etc., then summarize its pipeline.

3. If the pipeline contains technical details, such as extensive feature engineering, hyperparameter tuning, etc., then list them in full detail.

4. Select a representative code segment for each pipeline. You must include dataset reading / submission generation parts. If task-specific details such as feature engineering are included, the code segment should contain them as well.

**Public Kernels**    {*contents of public kernels*}

---

**Response Template of Strategy Distillation of Public Kernels**

**Pipelines**    Description of each strategy, separated by ===SEPARATOR=== mark. For each strategy, follow this format:

- Pipeline: A full detailed description of the pipeline. All input/output format, hyperparameters, training settings, model architectures, feature engineering, validation metric, and any other relevant information should be included. **Do not omit any feature engineering details**.
- Code abstract: A representative code segments that captures the essence (including input/output) and novelty of the pipeline. You **MUST** go through all the publicly available code and **include the parts that generate the submission file**. Contain task-specific engineering details. Mark the remainder as ellipses.

---

**Prompt for Strategy Distillation of Public Discussions**

**Introduction**    You are an expert machine learning researcher preparing for the Kaggle competition described below.

**Task Description**    {*description of the specified task*}

**Goals**    These are top-voted public discussions during the competition. Your job is to:

**Public Discussions**    {*contents of public discussions*}

1. Carefully read the following discussions.

2. For each discussion, you should decompose it into critical, novel and inspiring ideas that have potential to win this competition.

---

**Response Template of Strategy Distillation of Public Discussions**

**Ideas**    required format: python list of strings, each element is a description of an idea extracted from the discussions. e.g. ['idea 1', 'idea 2'].

---

## I.3    IDEA PROPOSER

---

**Prompt for Brainstorm**

**Introduction**    You are an expert machine learning researcher preparing for the Kaggle competition described below.

**Task Description**    {*description of the specified task*}

**Goals**    I already have a list of ideas that partially explore how to approach this competition. Your job is to:

1. Think creatively and construct at least **4 alternative and highly novel solution paths** that are likely to perform well, especially if combined with careful experimentation.

2. Each solution path can be a strategy, pipeline, or method that combines multiple techniques. Try to make them as different as possible from the existing "ideas" list.

---

3. After describing each full solution path, **break it down into individual minimal ideas**- these should be the smallest units of implementation (e.g., "use LightGBM for baseline", "normalize input features", "apply stratified K-fold CV")

4. Ensure these ideas do not substantially duplicate items already in "ideas".

5. Refer to the "Reports" section for the latest updates and suggestions on the ideas and previous pipelines.

**Ideas**   {*entries in the idea pool*}

**Reports**   {*entries in the report pool*}

**Public Pipelines**   {*all public pipelines extracted before*}

**Instructions**   Format your output like this (one line, one idea):
{*your understanding of the task and explanation of your approaches*}
===SOLUTION_PATH_1===
{*description of this approach*}
- {*minimal idea 1*}
- {*minimal idea 2*}
- {*minimal idea 3*}
- ...
===SOLUTION_PATH_2===
...
===SOLUTION_PATH_3===
...
Be ambitious but realistic - many ideas can later be tested on a small subset of the data. Focus on novelty, diversity, and decomposability. Ready? Start.

---

**Prompt for Idea Filtering and Reconstruction**

**Introduction**   You are a machine learning expert. After carefully searching the relevant literature, you have come up with a list of ideas to implement. However, this idea list has some issues:
• Some ideas are too similar and should be merged into one.
• Some ideas are overlapping, you should rephrase and decouple them.
• You should discard ideas that are irrelevant to the final performance, such as error visualization, etc.
  You should refer to the Reports section and Public Pipelines section for previous implemented pipelines. Please decompose, merge, and reconstruct the ideas listed below.

**Ideas**   {*entries of the idea pool*}

**Reports**   {*entries of the report pool*}

**Public Pipelines**   {*all public pipelines extracted before*}

---

**Response Template of Idea Filtering and Reconstruction**

**Ideas**   required format: Python list of strings, each element is a description of an idea. e.g. ['idea 1', 'idea 2'].

---

**Prompt for Coding Agent Report Compilation**

Please summarize the results and submit a comprehensive report.

---

**Response Template for Coding Agent Report Compilation**

**pipeline**   A detailed description of the pipeline that generated the best results. All hyperparameters, training settings, model architectures, feature engineering, validation metric, and any other relevant information should be included. Describe potential improvements and future work.
**summary**   A comprehensive evaluation of each individual component of the pipeline. For each component, summarize in the following format:
=== {*name of the component*} ===

**Novelty**: 0-10 (0: trivial, 10: clearly novel - major differences from existing well-known methods)
{*your rationale*}
**Feasibility**: 0-10 (0: almost impossible to implement and require extensive engineering, 10: Easy to implement)
{*your rationale*}
**Effectiveness**: 0-10 (0: minimal performance improvement, 10: very strong performance, significantly outperform most baselines)
{*your rationale*}
**Efficiency**: 0-10 (0: very slow, over-dependent on CPU and hard to produce meaningful results within the time limit, 10: high utilization of GPU)
{*your rationale*}
**Confidence**: 0-10 (0: no emprical results, not sure whether the evaluation is correct, 10: fully verified on large scale with abundant results)

## I.4 EVALUATOR

### Prompts for Dataset Splitting and evaluate.py

You are an experienced machine learning engineer. Please generate two self-contained Python code for local evaluation of a Kaggle agent. Your code should be robust, reusable, accept command-line arguments and print necessary information.

**Background**
- Kaggle competitions usually provide labels only for the training set. To evaluate an agent locally, we need to split the training set into a training and validation split.
- The validation set must hide its labels from the agent. The agent only sees the training set (with labels) and the validation inputs (without labels).
- The hidden validation labels will be stored separately and used only for offline evaluation.
- Importantly: ./public must never contain validation labels. Validation labels are saved only in ./private.

**Kaggle Competition Description**  {*description of the specified task*}

**Data Preview**  {*schema of the input file structure*}

**Deliverables**  Please generate two scripts (both in Python 3, runnable from the command line):
**1) split_dataset.py**
**Goal**: Split the original training data into 90% training and 10% validation. Store validation inputs (without labels) in ./public, and validation labels in ./private. The training set (with labels) and original test set must remain in ./public, preserving the original structure as closely as possible. The structure of validation inputs should also match the test set. Generate a sample validate submission validate_sample_submission.csv under ./public. All original data (training and test) are visible in {*path to the input directory*}.
**Example**: If the original data is structured as:

```
- kaggle_evaluation/ (official evaluation tool provided by Kaggle)
   - __init__.py
   - ...
- train.csv
- train/
- test.csv
- test/
- sample_submission.csv
```

You should split the dataset into:

```
(./public/)
- kaggle\_evaluation/ (official evaluation tool provided by Kaggle) (unchanged, soft
    links)
   - __init__.py
   - ...
- train.csv (this contains 90% of the training data)
- train/ (this contains 90% of the training data, keep unchanged data as soft links)
- test.csv (unchanged, soft link)
- test/ (unchanged, soft link)
- sample_submission.csv (unchanged, soft link)
- validate.csv (this contains 10% of the training data with labels withheld)
- validate/ (soft links)
- validate_sample_submission.csv (a sample submission file for validation set)
```

```
(./private/)
- validate.csv (labels of validation set)
```

If the training data contains zip files, you should extract them to the public directory before splitting the dataset. You should always print the directory structure after the split. Do not extract files to the original directory and keep it unchanged.

If the training data contains multiple classes, you should use **stratified sampling**. You should strictly follow the evaluation metric mentioned in the task description and ensure the validation set is representative of the overall class distribution. Never write validation labels into ./public.

Your code will be executed by command line as follows:

```bash
python split_dataset.py --input_dir <path to the input directory> --public_dir ./public
    --private_dir ./private
```

DO NOT store the training and test files in other folders such as ./public_¡TIMESTAMP¿, the ./public folder will be exposed to later code agent. Make sure the ./public directory has similar structure with the original data folder.

**2) evaluate.py**

Goal: Evaluate the agent's predictions on validation set against the hidden ground truth (./private/...). Output evaluation results (json format) to console and write ./private/eval_report.json.

It will be executed by command line as follows:

```bash
python evaluate.py --public\_dir ./public --private\_dir ./private --pred <path to the
    validation submission file>
```

We will pass the path to the sample validation submission file as the argument to your evaluate.py script. It typically produces low scores.

The script should generate in the following json format at ./private/eval_report.json:

```
{
   "score": A float number represents the evaluation score on the validation set. Do
       not omit this field. If the evaluation is unsuccessful or the predictions are
       invalid, this field should be set to null,
   "success": A boolean value indicates whether the evaluation was successful or not,
   "message": A string provides additional information about the evaluation result.
       Leave it an empty string if the predictions are valid and evaluation is
       successful. Otherwise provide necessary details on why it failed.
}
```

Do not raise any error or exception. If the evaluation is unsuccessful, you should set the score to null and provide a detailed explanation in the message field.

Now, let's write these two scripts step by step. Your should first generate split_dataset.py. We will execute the code by command line as mentioned above. You should correct the code in case of any issues. You should always generate full, self-contained code. No part of the code should be omitted. Respond in the following format:

```current_file
This should be either split_dataset.py or evaluate.py. Leave this as None if both are
    generated and functioned. This indicates the current file you are editing.
```

```explanation
You explanation on the workflow of your code.
```

```python
The full content of the current file. Leave this as None if both are generated and
    functioned.
```

## I.5 Coding Agent

---

**Prompts for Coding Agent Iterative Implementation**

**Introduction**  You're an expert Kaggle competitor tasked with implementing a pipeline into Python code. You can modify the details (training parameters, feature engineering, model selection, etc. ), but do not change overall architecture of this pipeline. The goal is to **obtain best score** on this competition.

**Task Description**  {*description of the specified task*}

**Pipeline**  {*description of the solution draft to implement*}

**Data Overview**  {*schema of the input file structure*} Follow the pipeline description and the code abstract to implement it. All the input files are visible in ../input folder, this folder typically contains the competition data and external resouces, including public datasets, models and outputs of other kernels. DO NOT USE /kaggle/input paths in your code. USE ../input instead.

file structure:

```
- input/ (../input)
    - competition_id/ # the official competition dataset
    - alice/dataset1/ # other public datasets
    - alice/kernel1/ # referenced kernels
- working/
    - agent.ipynb # the notebook you will be working on (./agent.ipynb)
    - other files
```

You will develop the pipeline based on this codebase. Any output files of the codebase, such as csvs, checkpoints, etc., are visible in ./, which is also your current working directory.

{*Description of Selected Codebase*}

You should note that checkpoints generated by this codebase is store in ./ other than ../input. You must load the checkpoint file under the ./ directory for ensemble prediction.

Your code must produce a submission at ./submission.csv, this is EXTREMELY IMPORTANT. Before generating the submission, you must print the value of the evaluation metric computed on a hold-out validation set. You can use custom evaluation functions during training, but the final metric **MUST FOLLOW THE EVALUATION SECTION IN THE TASK DESCRIPTION** on a validation set. If other kernels with submission.csv are provided in the input folder, you can ensemble them before generating your own submission. This is important because we will pick your best code based on this metric. You are allowed to load the checkpoints of other models. Do not contain any absolute paths in your code. Time limit per run is 2 hours. Your code will be killed if timeout.

Your code will be executed on a single A6000 GPU. Use large batchsizes to maximize the gpu utilization. If the code segment is provided in this prompt, you should follow the input/output structure. You are allowed to install any packages you need or inspect the workspace (e.g., print file contents, check folder structure). Always use gpu for acceleration. DO NOT USE ABSOLUTE PATHS IN YOUR CODE.

The workspace will be maintained across iterations. That is, if your first iteration code produces a checkpoint, you can load it in the second iteration. You can ensemble submissions generated by yourself and other kernels. You should generate model checkpoints for future loading. If you load the external submissions successfully but failed to merge them with your own predictions, you should print the headers of the external submission and your own predictions and check if the ids are aligned. All the external submissions are valid. Your predictions should be in the same format as them.

To evaluate your submission locally. You should also generate a submission file on the validation set. All the validation data are typically structured similarly to the test data. An external grader will be used to evaluate your validation submission. That is to say, you should generate TWO submission files: one is for the validation set and the other is for the test set. Generate two submission files in the same code cell.

You are allowed to install any packages by running 'pip install ¡package_name¿' in your script. Your installation will take effect in the NEXT cell.

A persistent Jupyter Notebook session is maintained. Your proposed code cell will be directly appended to the notebook and executed. You should separate data loading, training and evaluation in different cells. Now, please propose THE FIRST CELL of your code (not your full code!) using the following format:

```
<goal>
The explanation of your first cell. You should describe the desired execution time and
    output of this cell. Explain how to interpret the execution output.
</goal>
```

```

The content of this cell. Do not wrap the code in a markdown block. Your code will be
    appended to the notebook, which is stored at ./agent.ipynb. Your code must print
    necessary information after each milestone.
```

```


<validation_submission>
The name of the submission file for the validation set. e.g. validate\_submission.csv.
    If your current code cell does not produce two submission files, leave this as
    None.
</validation_submission>

<submission>
The name of the submission file for the test set. e.g. submission.csv. This submission
    should be ready for Kaggle submission. If your current code cell does not produce
    two submission files, leave this as None.
</submission>
```

The validation_submission tag and the submission tag should must be both empty or both non-empty.

**Prompt for Execution Monitor**

You are an AI assistant monitoring code execution. Your task is to analyze the current execution output and decide whether the code should continue running.
Code being executed:
{*code to analyze*}
Goal: {*execution target of this code*}
Runtime Information:
- Current runtime: {*code execution time elapsed*}
- Maximum runtime: {*maximum execution time*}
- Remaining time: {*remaining execution time*}
Current Output:
{current output of this code cell}
Consider these factors:

1. Is the loss exploding (becoming very large or NaN)?

2. Is the loss decreasing normally over time?

3. Are there any error messages indicating failure?

4. Does the output suggest normal training/execution progress?

5. Based on current progress and remaining time, is it possible to complete within the time limit?

Respond in the following format:

```
<action>
CONTINUE/STOP
</action>

<explanation>
Your rationale for the action. Describe the current progress, your estimated remaining
    time, and explain why you think the execution should continue or stop. DO NOT GIVE
     SUGGESTIONS ON BUG FIXES.
</explanation>
```

**Prompt for Consequent Code Revisions**

The execution takes {*execution_time*} seconds and ends with the following output:
{*truncated output*}
Execution completed successfully. You should keep updating your code (e.g., try different hyperparameters, augmentations, model architectures) after you have made successful submission. Your best submission will be recorded.
Now, respond in the following format:

```
<validation_submission>
The name of the submission file for the validation set. e.g. validate_submission.csv.
    If your current code cell does not produce a submission file on the validation set
    , leave this as None.
</validation_submission>

<submission>
```

```
The name of the submission file for the test set. e.g. submission.csv. This submission
    should be ready for Kaggle submission. If your current code cell does not produce
    a submission file on the test set, leave this as None.
</submission>

<goal>Describe the goal and how to inspect the output of your next code cell</goal>


The content of your next code cell. Following the previous format, do not wrap your
    code within markdown code marks. You should keep updating your code (e.g., try
    different hyperparameters, augmentations, model architectures) even after you have
     made successful submission. Always evaluate your submission and print the metric
    on a validation set.

```

The validation_submission tag and the submission tag should must be both empty or both non-empty.

## J  CASE STUDY: DENOISING DIRTY DOCUMENTS

### J.1  DATASET PREPARATION

Besides the task description and datasets prepared in MLE-Bench, MLE-Live collects 59 public kernels and 19 discussions which are available on Kaggle and are posted before the competition ends.

### J.1.1  EXAMPLE OF PUBLIC KERNEL

```python
"""
A simple feed-forward neural network that denoises one pixel at a time
"""
import numpy as np
import theano
import theano.tensor as T
import cv2
import os
import itertools

theano.config.floatX = 'float32'

def load_image(path):
    return cv2.imread(path, cv2.IMREAD_GRAYSCALE)

def feature_matrix(img):
    """Converts a grayscale image to a feature matrix

    The output value has shape (<number of pixels>, <number of features>)
    """
    # select all the pixels in a square around the target pixel as
        features
    window = (5, 5)
    nbrs = [cv2.getRectSubPix(img, window, (y, x)).ravel()
            for x, y in itertools.product(range(img.shape[0]), range(img.
                shape[1]))]

    # add some more possibly relevant numbers as features
    median5 = cv2.medianBlur(img, 5).ravel()
    median25 = cv2.medianBlur(img, 25).ravel()
    grad = np.abs(cv2.Sobel(img, cv2.CV_16S, 1, 1, ksize=3).ravel())
    div = np.abs(cv2.Sobel(img, cv2.CV_16S, 2, 2, ksize=3).ravel())

... (omitted) ...

    # for fname in os.listdir('../input/test/'):
    for fname in ['1.png']:
        test_image = load_image(os.path.join('../input/test', fname))
        test_x = feature_matrix(test_image)

        y_pred, = predict(test_x)
        output = y_pred.reshape(test_image.shape)*255.0

        cv2.imwrite('original_' + fname, test_image)
        cv2.imwrite('cleaned_' + fname, output)

if __name__ == '__main__':
    main()
```

### J.1.2  EXAMPLE OF DISCUSSION

```
# Edge Diffraction in train_cleaned data
(Lance <TIER: N/A>) <p>I'm studying the pixels in train_cleaned data.  I attached a
    colorized blow-up version of part of the image train_cleaned/45.png.   The
```

```
        yellow pixels are any pixels that were not pure white ( != 0xFF gray scale) in image 45.
        png, the green was pure white (0xFF).</p>
    <p>So you see what looks like an edge diffraction line lining the outer edge of all the
        letters.</p>
    <p>Okay, maybe I got something wrong in my code.  Can anyone confirm this edge
        diffraction thing in the train_cleaned data, as for example the first word in
        train_cleaned/45.png (There).  You need to make the non-white (byte != 0xFF)
        pixels all a more contrasting color or you may not see it.</p>
    <p>I'm guessing that the clean png files were at some point scanned in using some kind of
        optical scanning machine which added these edge diffraction lines when the light
        diffracts off the edge of the black ink character.</p>
    ... (omitted) ...
    + (Rangel Dokov <TIER: MASTER>) <p>Yes, there is some noise, which doesn't look like it
        should be there in the clean set... I ran a test setting everything whiter that 0xF5
        to 0xFF and the RMSE was 0.005, which should be an upper bound on the effects from the
        halos. This will likely be large enough to make the top of the leaderboard a game of
        luck, but since this is just a playground competition I'm not terribly worried about
        it.</p>
```

## J.2 EXAMPLE AGENT WORKFLOW

In our experiment settings, CoMind only accesses top-10 voted discussions and kernels and ignores the rest.
The community is initialized with these artifacts. Upon completion of this process, 7 ideas and 10 pipelines are
generated. Below is an excerpt of the ideas and reports generated by the Analyzer.

```
(0) Use behaviour-based clustering of neural networks: cluster models by their error patterns
     and ensemble them for document enhancement
(1) Implement sliding-window patch-based models that take an input window and output multiple
     cleaned pixels simultaneously for both denoising and resolution enhancement
(2) Apply a Waifu2x-inspired deep convolutional neural network with gradually increasing
     filter counts (e.g., 1 -> 32 -> 64 -> 128 -> 256 -> 512 -> 1) and LeakyReLU activations
     for effective denoising
(3) Carefully initialize convolutional weights (e.g., stdv = sqrt(2/(kW*kH*nOutputPlane))) and
     use LeakyReLU to improve model convergence and performance
(4) Ensemble multiple models with different input preprocessing: combine outputs from a pure
     CNN, background-removed images, edge maps, and thresholded inputs to capture diverse
     noise characteristics
(5) Augment training data to simulate real-world 3D deformations and shadows on text, not just
     2D noise, to better match test-time artifacts
(6) Account for systematic artifacts in 'clean' training data (e.g., single-pixel halos) by
     treating them as noise or adjusting targets accordingly during training
```

```
Public pipeline (0): - Pipeline: A simple feed-forward neural network that denoises one pixel
     at a time (Theano).
  - Feature engineering: for each pixel extract a 5*5 window of gray values (neighbors), 5*5
      median blur, 25*25 median blur, Sobel gradient and second-order derivative magnitudes,
      stack into a feature vector. Normalize features to [0,1].
  - Model architecture: two-layer MLP; hidden layer size N_HIDDEN=10, tanh activation, output
      layer with custom activation clip(x+0.5,0,1).
  - Training: MSE cost, stochastic gradient descent with learning rate 0.1, batch size 20,
      epochs 100. Validation on one image (3.png) at each epoch by RMSE.
  - Prediction: apply same feature_matrix to test images, predict pixel values, reshape to full
       image, write out cleaned PNGs.
- Code abstract:
   def feature_matrix(img):
       window=(5,5)
       nbrs=[cv2.getRectSubPix(img,window,(y,x)).ravel()
            for x,y in itertools.product(range(img.shape[0]),range(img.shape[1]))]
       median5=cv2.medianBlur(img,5).ravel()
       median25=cv2.medianBlur(img,25).ravel()
       grad=np.abs(cv2.Sobel(img,cv2.CV_16S,1,1,ksize=3).ravel())
       div=np.abs(cv2.Sobel(img,cv2.CV_16S,2,2,ksize=3).ravel())
       misc=np.vstack((median5,median25,grad,div)).T
       features=np.hstack((nbrs,misc))
       return (features/255.).astype('float32')
   ...
   class Model(object):
       def __init__(...):
          self.layer1=Layer(...,n_in=...,n_out=N_HIDDEN,activation=T.tanh)
          self.layer2=Layer(...,n_in=N_HIDDEN,n_out=n_out,
                        activation=lambda x: T.clip(x+0.5,0,1))
       def cost(self,y): return T.mean((self.output-y)**2)
   ...
---------- PIPELINE SEPARATOR ----------
Public pipeline (1): - Pipeline: Matching image backgrounds in R (no ML model).
  - Reads test PNGs in batches of 12 images.
  - Flattens each into vectors of size 258*540, stacks as columns.
  - For each pixel location, takes the maximum value across images as an estimate of background
      .
```

```
  - Writes out background images as PNG.
- Code abstract:
  for(i in 1:4) {
    matches=seq(1,205,by=12)+(i-1)*3
    rawData=matrix(0,258*540,length(matches))
    for(j in seq_along(matches)){
      imgY=readPNG(file.path(testDir,paste0(matches[j],'.png')))
      rawData[,j]=as.vector(imgY[1:258,1:540])
    }
    background=matrix(apply(rawData,1,max),258,540)
    writePNG(background, paste0('background',matches[j],'.png'))
  }
  ...
---------- PIPELINE SEPARATOR ----------
Public pipeline (2): - Pipeline: Pixel-wise Random Forest regression (Python, chunk size=1e6).
  - Feature engineering: pad image by mean value (padding=1); extract 3*3 neighborhood per
      pixel, flatten as features.
  - Training data: load all train noisy images, compute features via joblib parallel (n_jobs
      =128), load targets as flattened clean pixel intensities/255.
  - Model: sklearn.ensemble.RandomForestRegressor(warm_start=True, n_jobs=-1). Incrementally
      add one estimator at a time: split training rows into CHUNKSIZE=1e6 slices, in each
      slice increase n_estimators by 1 and fit on that slice.
  - Prediction: extract test features similarly, generate index strings "image_row_col",
      predict pixel values, write submission CSV.
- Code abstract:
  def get_padded(img, padding=1):
    padval=int(round(img.mean()))
    ... return padded
  def get_features_for_image(img,padding=1):
    padded=get_padded(img,padding)
    return np.vstack([padded[i:i+3,j:j+3].reshape(1,-1)
                   for i in range(rows) for j in range(cols)])
  ...
  def get_model(X,y):
    model=RandomForestRegressor(n_estimators=0,warm_start=True,n_jobs=-1)
    for start,end in slices:
        model.set_params(n_estimators=model.n_estimators+1)
        model.fit(X[start:end],y[start:end])
    return model
  ...
---------- PIPELINE SEPARATOR ----------
... (omitted) ...
---------- PIPELINE SEPARATOR ----------
Public pipeline (9): - Pipeline: XGBoost with basic features in R.
  - Features: raw pixel value, kmeansThreshold binary, background_Removal feature.
  - Sample 25k pixels, build DMatrix, 5-fold CV, early stop, select rounds, train xgb.
  - Predict on test, clip, build id strings, write submission.csv.
- Code abstract:
  dat=cbind(y,raw=img2vec(imgX),thresholded=kmeansThreshold(imgX),
         backgroundRemoval=img2vec(background_Removal(imgX)))
  rows=sample(nrow(dat),25000)
  dtrain=xgb.DMatrix(dat[rows,-1],label=dat[rows,1])
  xgb.tab=xgb.cv(data=dtrain,nfold=5,early.stop.round=50)
  xgb.mod=xgboost(data=dtrain,nrounds=min.err.idx)
  yHat=predict(xgb.mod,newdata=as.matrix(x))
  ...
```

Then, the Idea Proposer begins brainstorming to propose more original strategies.

```
===SOLUTION_PATH_1===
Multi-Scale Wavelet-U-Net with Attention and Multi-Term Loss
Description: decompose each image via discrete wavelet transform (DWT) into low-/high-
    frequency bands, process each band in a dual-branch U-Net enhanced with channel- and
    spatial-attention, then invert the wavelet to reconstruct. Train end-to-end with a
    weighted sum of pixel-wise RMSE, wavelet-band consistency, and edge preservation losses.
- apply 2D DWT to noisy input -> obtain LL, LH, HL, HH sub-bands
- feed LL into a "coarse" encoder branch, feed concatenated LH/HL/HH into a "detail" encoder
    branch
- use a U-Net decoder to upsample each branch back to patch size, fuse via learned 1*1
    convolutions
- insert Convolutional Block Attention Modules (CBAM) after each encoder and decoder block
- define loss = alpha*pixelRMSE(clean,output) + beta*bandRMSE(wavelet(clean),wavelet(output))
    + gamma*edgeLoss(Sobel(clean),Sobel(output))
- train on full images with AdamW and a cosine-annealing LR schedule

===SOLUTION_PATH_2===
Stroke-Aware Conditional GAN with OCR-Guided Perceptual Loss
Description: build a conditional GAN (generator = deep residual encoder-decoder, discriminator
    = PatchGAN) that not only minimizes pixel loss but also preserves text strokes-enforce a
    stroke-level loss via a pre-trained small CNN classifier that predicts presence/width of
```

```
          strokes. Add an OCR-based perceptual loss: feed predictions through a frozen OCR engine
          embedding and minimize distance to clean embedding.
- implement generator as ResNet blocks + skip connections (64->128->256->128->64)
- implement discriminator as 70*70 PatchGAN to focus on local texture
- include L1 pixel loss + adversarial loss + stroke consistency loss (L1 between stroke-CNN
          features on clean vs. restored)
- freeze a small text-structure CNN (trained on binary masks) to extract stroke features
- run Tesseract (or lightweight OCR CNN) on restored vs. clean, extract penultimate-layer
          activations, add perceptual loss term
- train with R1 gradient penalty and spectral normalization on discriminator

===SOLUTION_PATH_3===
Joint Dictionary Learning + Non-Local Patch Aggregation
Description: learn paired dictionaries (D_noisy, D_clean) for small patches (e.g. 8*8) via
          coupled K-SVD. At test time, extract overlapping patches, compute sparse codes alpha
          under D_noisy via OMP, reconstruct clean patches = D_clean*alpha. Then, perform non-local
          means on the reconstructed patches to exploit self-similarity and average aggregates.
- sample a large bank of noisy/clean patch pairs, initialize D_noisy, D_clean with DCT basis
- run coupled K-SVD to minimize ||D_noisy*alpha - y_noisy|| + ||D_clean*alpha - y_clean|| w.r.
          t. D_noisy,D_clean,alpha
- at test time, for each image patch y_noisy, compute alpha via Orthogonal Matching Pursuit (
          sparsity <= k)
- reconstruct y_clean_est = D_clean*alpha for each patch
- perform block-matching to find K nearest patches per reference patch (Euclid dist), stack
          them
- aggregate reconstructed patches with non-local weights (e.g. Gaussian on reconstruction
          residual)

===SOLUTION_PATH_4===
Self-Supervised Blind Denoising via Noise2Void + Test-Time Adaptation
Description: exploit purely noisy data-train a small U-Net with masked pixel prediction (
          Noise2Void) on each test image at inference (test-time training). The network learns to
          predict a pixel from its context, gradually adapting to local noise statistics, then you
          run a forward pass to obtain the cleaned image. No clean target needed.
- define blind-spot or random masking scheme: mask 1% pixels per batch, replace with neighbors
- build a lightweight CNN (e.g. 5 down/up blocks with skip connections) that predicts a full
          image
- fine-tune this CNN on each test image for N_iter (e.g. 500 steps) using only masked L2 loss
- use data augmentation (rotations, flips) on the single test image to diversify contexts
- after adaptation, perform a clean forward pass without masking to get the denoised output
- optionally ensemble outputs from multiple random initializations to reduce variance
```

To remove similar ideas and decompose overlapped ideas, a reconstruction is performed subsequently. 9 ideas are preserved after the filtering and reconstruction. These ideas are then merged with the idea memory.

```
(0) Use diverse ensembles by clustering models based on their error patterns and combining
          outputs from differently preprocessed inputs (e.g., raw, background-removed, edge maps,
          thresholded) to capture varied noise characteristics.
(1) Develop a sliding-window, patch-based model that takes an input region and predicts
          multiple denoised and super-resolved pixels simultaneously for efficient document
          enhancement.
(2) Implement a Waifu2x-inspired deep convolutional network with progressively increasing
          filter counts and LeakyReLU activations, initialized using He normalization for robust
          convergence in denoising tasks.
(3) Augment the training set with simulated 3D deformations, lighting variations, and shadows
          on text to better match real-world scanning artifacts.
(4) Identify and compensate for systematic artifacts in the "clean" training targets (e.g.,
          single-pixel halos) by modeling them as noise or adjusting ground-truth labels during
          training.
(5) Build a wavelet-domain dual-branch U-Net: decompose inputs with 2D DWT into approximation
          and detail bands, encode separately, decode and fuse via learned 1*1 convolutions,
          integrate CBAM attention modules, and train end-to-end with a composite loss combining
          pixel RMSE, wavelet-band RMSE, and edge loss, optimized using AdamW with a cosine-
          annealing LR schedule.
(6) Apply a GAN-based restoration framework: use a ResNet-based generator and 70*70 PatchGAN
          discriminator, train with a combination of L1 pixel loss, adversarial loss, stroke-
          consistency loss (via a frozen stroke-feature CNN), and perceptual OCR-feature loss,
          incorporating R1 gradient penalty and spectral normalization.
(7) Leverage coupled dictionary learning and non-local aggregation: learn noisy and clean
          dictionaries simultaneously via coupled K-SVD, reconstruct patches using OMP, enhance
          self-similarity with block-matching, and fuse reconstructed patches with non-local
          weighting.
(8) Implement blind-spot test-time adaptation: randomly mask a small percentage of pixels per
          batch, train a lightweight CNN with skip connections on the single test image using
          masked L2 loss and augmentations, fine-tune per image for a few hundred steps, and
          optionally ensemble multiple initializations for variance reduction.
```

Based on the ideas received from the Idea Proposer and previous reports, the Coordinator then compiles 2 solution drafts.

```
Pipeline 1 (Baseline CNN - U-Net with L1 loss and simple augmentations)
-----------------------------------------------------------------------------
Overview
- A lightweight U-Net trained on 256*256 patches of noisy/clean document pairs.
- Single-branch encoder-decoder with skip connections, ReLU activations, batch norm.
- Loss: L1 pixel-wise loss (less sensitive to outliers than MSE), optional total variation
     regularization.
- Optimizer: Adam; train on 1 A6000 within 1 hr.

Data Preparation
1. Read all noisy (input) and clean (target) train images, normalize intensities to [0,1].
2. Extract random 256*256 patches (stride = 128) with matching noisy/clean pairs.
3. Data augmentation: random horizontal/vertical flips, +-90 degree rotations.
4. Create PyTorch DataLoader with batch_size=16 (fits 48 GB) for ~100 k patches.

Network Architecture (PyTorch pseudocode)
```
```
class UNet(nn.Module):
    def __init__(self):
        super().__init__()
        # Encoder
        self.enc1 = DoubleConv(1, 64)
        self.enc2 = Down(64,128)
        self.enc3 = Down(128,256)
        self.enc4 = Down(256,512)
        # Bottleneck
        self.bottleneck = Down(512,512)
        # Decoder
        self.up4 = Up(1024,256)
        self.up3 = Up(512,128)
        self.up2 = Up(256,64)
        self.up1 = Up(128,64)
        self.final = nn.Conv2d(64,1,kernel_size=1)
    def forward(self,x):
        e1=self.enc1(x)
        e2=self.enc2(e1)
        e3=self.enc3(e2)
        e4=self.enc4(e3)
        b = self.bottleneck(e4)
        d4=self.up4(b,e4)
        d3=self.up3(d4,e3)
        d2=self.up2(d3,e2)
        d1=self.up1(d2,e1)
        return torch.sigmoid(self.final(d1))
```
```
Helper modules:
- DoubleConv = (Conv2d -> BatchNorm2d -> ReLU) *2
- Down = (MaxPool2d -> DoubleConv)
- Up = (ConvTranspose2d for upsampling -> concatenate skip -> DoubleConv)

Training
- Loss = L1Loss(output, target) + lambda*TV(output) (lambda=1e-5 for smoothness).
- Optimizer = Adam(lr=1e-3, weight_decay=1e-5).
- LR schedule: ReduceLROnPlateau(monitor=val_loss, factor=0.5, patience=5).
- Train for up to 50 epochs; early-stop if val_loss stagnates.
- Validation: hold out 10% patches to monitor RMSE.

Inference
- For each test image (e.g., 540*258), slide 256*256 window with stride=128, predict, and
     average overlapping outputs.
- Threshold nothing; output raw [0,1] floats per pixel.

Compute Budget
- ~100 k patches, batch 16, ~6 k steps per epoch. On A6000: ~2-3 min/epoch => 50 epochs ~ 2 hr
     ; with early stopping < 1 hr.

Pipeline 2 (Advanced Wavelet U-Net with CBAM and Composite Loss)
-------------------------------------------------------------------
Overview
- Dual-branch U-Net operating in wavelet domain (Haar DWT) to explicitly denoise tonal and
     textural components.
- CBAM (Convolutional Block Attention Modules) to adaptively weigh spatial/channel features.
- Loss = alpha*L1_pixel + beta*L2_wavelet + gamma*EdgeLoss.
- Optimizer = AdamW + CosineAnnealingLR.

Data Preparation
- Same as Pipeline 1 (256*256 patches + augmentations).
- On-the-fly DWT: for each noisy patch, compute one-level Haar DWT -> yields approximation (A)
       and details (H,V,D).

Network Architecture
```

```
(implemented in PyTorch)
```
class WaveletUNet(nn.Module):
    def __init__(self):
        super().__init__()
        # Shared CBAM-Res blocks for Approx and Detail branches
        self.encA1 = CBAMResBlock(1,64)
        self.encD1 = CBAMResBlock(3,64)
        self.pool = nn.MaxPool2d(2)
        self.encA2 = CBAMResBlock(64,128)
        self.encD2 = CBAMResBlock(64,128)
        # Bottleneck
        self.bottleneck = CBAMResBlock(256,256)
        # Decoder
        self.up2 = UpRes(256,128)
        self.up1 = UpRes(128,64)
        self.final = nn.Conv2d(64,4,kernel_size=1)
    def forward(self, x):
        # x shape: [B,1,H,W]
        A, (H,V,D) = haar_dwt(x) # PyWavelets or custom conv
        d0 = torch.cat([self.encA1(A), self.encD1(torch.cat([H,V,D],dim=1))],dim=1)
        p1 = self.pool(d0)
        d1 = torch.cat([self.encA2(p1[:, :128]), self.encD2(p1[:, 128:])],dim=1)
        b = self.bottleneck(self.pool(d1))
        u2 = self.up2(b,d1)
        u1 = self.up1(u2,d0)
        out = self.final(u1) # [B,4,H,W]
        # Reconstruct denoised x via inverse DWT
        A_hat, H_hat,V_hat,D_hat = out[:,0:1], out[:,1:2], out[:,2:3], out[:,3:4]
        return ihaar_dwt((A_hat, (H_hat,V_hat,D_hat)))
```
Key building blocks:
- CBAMResBlock: residual block with Conv-BN-ReLU->Conv-BN, plus CBAM (channel & spatial
    attention after second Conv).
- UpRes: ConvTranspose2d upsample->concatenate skip->CBAMResBlock.

Loss
- L1_pixel = L1(output, target).
- L2_wavelet = L2(DWT(output) - DWT(target)).
- EdgeLoss = L1(Sobel(output), Sobel(target)).
- TotalLoss = alpha*L1_pixel + beta*L2_wavelet + gamma*EdgeLoss (alpha=1, beta=0.5, gamma=0.2)
    .

Training
- Optimizer: AdamW(lr=1e-3, weight_decay=1e-4).
- Scheduler: CosineAnnealingLR(T_max=50).
- Batch size=8 (higher memory due to dual branches).
- Epochs=50, early-stop by composite loss on val set.

Inference
- Same sliding-window approach as Pipeline 1.

Compute Budget
- CBAM and wavelet layers add ~1.5* compute but within A6000 48 GB: patch batch=8 yields ~250
    steps/epoch, so 50 epochs ~2 hr. Use early-stop or T_max=30 to finish within 1 hr.

References
- CBAM: Woo et al., "CBAM: Convolutional Block Attention Module" (ECCV 2018).
- Haar DWT: PyWavelets or custom conv filters.
- Edge loss: Sobel via fixed conv kernels.

Expected Benefits
- Explicit noise suppression in high-frequency bands.
- Attention modules focus on text strokes and edge restoration.
- Composite loss balances global structure and fine details.
```

In our experiment, 2 parallel Coding Agents are invoked to implement the solution drafts respectively. Each instance has a quota of 20 steps, 3 hours in total and 1 hour per execution. An LLM is instructed to monitor the execution process and terminate failed runs.

The best code snippet generated in the first iteration is:

```
1  import os, glob, random, math
2  from PIL import Image
3  import torch
4  import torch.nn as nn
5  import torch.nn.functional as F
6  from torch.utils.data import Dataset, DataLoader
```

```python
from torchvision import transforms

# -- 1) Haar DWT / inverse DWT --
class HaarDWT(nn.Module):
    def __init__(self):
        super().__init__()
        wt = torch.zeros(4, 1, 2, 2)
        wt[0, 0, :, :] = 0.5
        wt[1, 0, 0, 0], wt[1, 0, 0, 1], wt[1, 0, 1, 0], wt[1, 0, 1, 1] = (
            0.5,
            -0.5,
            0.5,
            -0.5,
        )
        wt[2, 0, 0, 0], wt[2, 0, 0, 1], wt[2, 0, 1, 0], wt[2, 0, 1, 1] = (
            0.5,
            0.5,
            -0.5,
            -0.5,
        )
        wt[3, 0, 0, 0], wt[3, 0, 0, 1], wt[3, 0, 1, 0], wt[3, 0, 1, 1] = (
            0.5,
            -0.5,
            -0.5,
            0.5,
        )
        self.register_buffer("weight", wt)

    def forward(self, x):
        return F.conv2d(x, self.weight, stride=2)

class HaarIDWT(nn.Module):
    def __init__(self):
        super().__init__()
        wt = HaarDWT().weight.clone()
        self.conv = nn.ConvTranspose2d(4, 1, 2, stride=2, bias=False)
        self.conv.weight.data.copy_(wt)
        self.conv.weight.requires_grad_(False)

    def forward(self, coeffs):
        return self.conv(coeffs)

# -- 2) Sobel edge for EdgeLoss --
class Sobel(nn.Module):
    def __init__(self):
        super().__init__()
        kx = torch.tensor(
            [[1, 0, -1], [2, 0, -2], [1, 0, -1]], dtype=torch.float32
        ).view(1, 1, 3, 3)
        ky = torch.tensor(
            [[1, 2, 1], [0, 0, 0], [-1, -2, -1]], dtype=torch.float32
        ).view(1, 1, 3, 3)
        self.register_buffer("wx", kx)
        self.register_buffer("wy", ky)

    def forward(self, x):
        gx = F.conv2d(x, self.wx, padding=1)
        gy = F.conv2d(x, self.wy, padding=1)
        return torch.sqrt(gx * gx + gy * gy + 1e-6)

# -- 3) CBAM, ResBlock, UpRes, WaveletUNet --
```

```python
72  class CBAM(nn.Module):
73      def __init__(self, c, r=16, k=7):
74          super().__init__()
75          self.mlp = nn.Sequential(
76              nn.Linear(c, c // r, bias=False),
77              nn.ReLU(inplace=True),
78              nn.Linear(c // r, c, bias=False),
79          )
80          self.spatial = nn.Conv2d(2, 1, kernel_size=k, padding=k // 2, bias=
                False)
81
82      def forward(self, x):
83          b, c, h, w = x.shape
84          avg = F.adaptive_avg_pool2d(x, 1).view(b, c)
85          mx = F.adaptive_max_pool2d(x, 1).view(b, c)
86          ca = torch.sigmoid(self.mlp(avg) + self.mlp(mx)).view(b, c, 1, 1)
87          x2 = x * ca
88          avgc = x2.mean(1, True)
89          maxc, _ = x2.max(1, True)
90          sa = torch.sigmoid(self.spatial(torch.cat([avgc, maxc], 1)))
91          return x2 * sa
92
93
94  class CBAMResBlock(nn.Module):
95      def __init__(self, inp, outp):
96          super().__init__()
97          self.conv1 = nn.Conv2d(inp, outp, 3, padding=1, bias=False)
98          self.bn1 = nn.BatchNorm2d(outp)
99          self.conv2 = nn.Conv2d(outp, outp, 3, padding=1, bias=False)
100         self.bn2 = nn.BatchNorm2d(outp)
101         self.cbam = CBAM(outp)
102         self.skip = (
103             nn.Conv2d(inp, outp, 1, bias=False) if inp != outp else nn.
                    Identity()
104         )
105         self.relu = nn.ReLU(inplace=True)
106
107     def forward(self, x):
108         idt = self.skip(x)
109         out = self.relu(self.bn1(self.conv1(x)))
110         out = self.bn2(self.conv2(out))
111         out = self.cbam(out) + idt
112         return self.relu(out)
113
114
115 class UpRes(nn.Module):
116     def __init__(self, ic, sc, oc):
117         super().__init__()
118         self.up = nn.ConvTranspose2d(ic, oc, 2, stride=2)
119         self.block = CBAMResBlock(oc + sc, oc)
120
121     def forward(self, x, skip):
122         x = self.up(x)
123         if x.shape[-2:] != skip.shape[-2:]:
124             x = F.interpolate(
125                 x, size=skip.shape[-2:], mode="bilinear", align_corners=False
126             )
127         return self.block(torch.cat([x, skip], 1))
128
129
130 class WaveletUNet(nn.Module):
131     def __init__(self):
132         super().__init__()
133         self.dwt = HaarDWT()
134         self.idwt = HaarIDWT()
```

```python
135         self.eA1 = CBAMResBlock(1, 64)
136         self.eD1 = CBAMResBlock(3, 64)
137         self.pool = nn.MaxPool2d(2)
138         self.eA2 = CBAMResBlock(64, 128)
139         self.eD2 = CBAMResBlock(64, 128)
140         self.b = CBAMResBlock(256, 256)
141         self.u2 = UpRes(256, 256, 128)
142         self.u1 = UpRes(128, 128, 64)
143         self.final = nn.Conv2d(64, 4, 1)
144
145     def forward(self, x):
146         A, H, V, D = self.dwt(x).chunk(4, 1)
147         a1 = self.eA1(A)
148         d1 = self.eD1(torch.cat([H, V, D], 1))
149         d0 = torch.cat([a1, d1], 1)
150         p1 = self.pool(d0)
151         pA, pD = p1[:, :64], p1[:, 64:]
152         a2 = self.eA2(pA)
153         d2 = self.eD2(pD)
154         d1b = torch.cat([a2, d2], 1)
155         b = self.b(self.pool(d1b))
156         u2 = self.u2(b, d1b)
157         u1 = self.u1(u2, d0)
158         out = self.final(u1)
159         return self.idwt(out)
160
161
162 # -- 4) OCRDataset for full training --
163 class OCRDataset(Dataset):
164     def __init__(self, noisy, clean, pp, ps, mode):
165         self.noisy = noisy
166         self.clean = clean
167         self.pp = pp
168         self.ps = ps
169         self.mode = mode
170         self.toT = transforms.ToTensor()
171
172     def __len__(self):
173         return len(self.noisy) * self.pp
174
175     def __getitem__(self, idx):
176         ii = idx // self.pp
177         n = Image.open(self.noisy[ii]).convert("L")
178         c = Image.open(self.clean[ii]).convert("L")
179         tn, tc = self.toT(n), self.toT(c)
180         H, W = tn.shape[-2], tn.shape[-1]
181         y = random.randint(0, H - self.ps)
182         x = random.randint(0, W - self.ps)
183         tn = tn[:, y : y + self.ps, x : x + self.ps]
184         tc = tc[:, y : y + self.ps, x : x + self.ps]
185         if self.mode == "train":
186             if random.random() < 0.5:
187                 tn, tc = torch.flip(tn, [-1]), torch.flip(tc, [-1])
188             if random.random() < 0.5:
189                 tn, tc = torch.flip(tn, [-2]), torch.flip(tc, [-2])
190             k = random.randint(0, 3)
191             if k > 0:
192                 tn, tc = torch.rot90(tn, k, [1, 2]), torch.rot90(tc, k, [1,
                         2])
193         return tn, tc
194
195
196 # -- 5) Prepare train/val split --
197 all_no = sorted(glob.glob("./input/train/*.png"))
198 all_cl = sorted(glob.glob("./input/train_cleaned/*.png"))
```

```
199  idxs = list(range(len(all_no)))
200  random.seed(42)
201  random.shuffle(idxs)
202  split = int(0.9 * len(idxs))
203  tr_idxs, va_idxs = idxs[:split], idxs[split:]
204  noisy_tr = [all_no[i] for i in tr_idxs]
205  clean_tr = [all_cl[i] for i in tr_idxs]
206  noisy_va = [all_no[i] for i in va_idxs]
207  clean_va = [all_cl[i] for i in va_idxs]
208
209  # -- 6) DataLoaders --
210  train_ds = OCRDataset(noisy_tr, clean_tr, pp=25, ps=256, mode="train")
211  val_ds = OCRDataset(noisy_va, clean_va, pp=1, ps=256, mode="val")
212  train_loader = DataLoader(
213      train_ds, batch_size=8, shuffle=True, num_workers=4, pin_memory=True
214  )
215  val_loader = DataLoader(
216      val_ds, batch_size=1, shuffle=False, num_workers=2, pin_memory=True
217  )
218
219  # -- 7) Model, optimizer, scheduler, loss w/ tuned weights --
220  device = torch.device("cuda" if torch.cuda.is_available() else "cpu")
221  model = WaveletUNet().to(device)
222  dwt = HaarDWT().to(device)
223  sobel = Sobel().to(device)
224  opt = torch.optim.AdamW(model.parameters(), lr=1e-3, weight_decay=1e-4)
225  sched = torch.optim.lr_scheduler.CosineAnnealingLR(opt, T_max=30)
226
227  # Tuned weights
228  alpha = 0.769634
229  beta = 0.430968
230  gamma = 0.250750
231
232
233  def composite_loss(o, t):
234      l1 = F.l1_loss(o, t)
235      l2 = F.mse_loss(dwt(o), dwt(t))
236      e1 = F.l1_loss(sobel(o), sobel(t))
237      return alpha * l1 + beta * l2 + gamma * e1
238
239
240  # -- 8) Train + validate --
241  best_rmse = 1e9
242  patience = 5
243  wait = 0
244  for epoch in range(1, 31):
245      model.train()
246      tloss = 0.0
247      for xb, yb in train_loader:
248          xb, yb = xb.to(device), yb.to(device)
249          pred = model(xb)
250          loss = composite_loss(pred, yb)
251          opt.zero_grad()
252          loss.backward()
253          opt.step()
254          tloss += loss.item()
255      sched.step()
256
257      # sliding-window validation
258      model.eval()
259      rmse_sum = 0.0
260      stride, ps = 128, 256
261      with torch.no_grad():
262          for nf, cf in zip(noisy_va, clean_va):
263              imn = Image.open(nf).convert("L")
```

```
264            imc = Image.open(cf).convert("L")
265            tn = transforms.ToTensor()(imn).unsqueeze(0).to(device)
266            tc = transforms.ToTensor()(imc).unsqueeze(0).to(device)
267            _, _, H, W = tn.shape
268            acc = torch.zeros_like(tn)
269            cnt = torch.zeros_like(tn)
270            xs = list(range(0, W - ps + 1, stride)) + [W - ps]
271            ys = list(range(0, H - ps + 1, stride)) + [H - ps]
272            for x in xs:
273                for y in ys:
274                    p = tn[:, :, y : y + ps, x : x + ps]
275                    out = model(p)
276                    acc[:, :, y : y + ps, x : x + ps] += out
277                    cnt[:, :, y : y + ps, x : x + ps] += 1
278            rec = acc / cnt
279            mse = F.mse_loss(rec, tc).item()
280            rmse_sum += math.sqrt(mse)
281        val_rmse = rmse_sum / len(noisy_va)
282        print(
283            f"Epoch {epoch}: train_loss={tloss/len(train_loader):.4f}, val_RMSE
                ={val_rmse:.6f}"
284        )
285
286        if val_rmse < best_rmse:
287            best_rmse = val_rmse
288            wait = 0
289            torch.save(model.state_dict(), "./working1/best.pth")
290        else:
291            wait += 1
292            if wait >= patience:
293                print("Early stopping.")
294                break
295
296    print("Best validation RMSE:", best_rmse)
297
298    # -- 9) Inference & submission --
299    model.load_state_dict(torch.load("./working1/best.pth"))
300    model.eval()
301    test_files = sorted(glob.glob("./input/test/*.png"))
302    with open("./working1/submission.csv", "w") as fout:
303        fout.write("id,value\n")
304        for tf in test_files:
305            im = Image.open(tf).convert("L")
306            t = transforms.ToTensor()(im).unsqueeze(0).to(device)
307            _, _, H, W = t.shape
308            acc = torch.zeros_like(t)
309            cnt = torch.zeros_like(t)
310            xs = list(range(0, W - ps + 1, stride)) + [W - ps]
311            ys = list(range(0, H - ps + 1, stride)) + [H - ps]
312            with torch.no_grad():
313                for x in xs:
314                    for y in ys:
315                        p = t[:, :, y : y + ps, x : x + ps]
316                        out = model(p)
317                        acc[:, :, y : y + ps, x : x + ps] += out
318                        cnt[:, :, y : y + ps, x : x + ps] += 1
319            rec = (acc / cnt).clamp(0, 1).cpu().numpy()[0, 0]
320            img_id = os.path.splitext(os.path.basename(tf))[0]
321            for r in range(H):
322                for c in range(W):
323                    fout.write(f"{img_id}_{r+1}_{c+1},{rec[r,c]:.6f}\n")
324
325    print("Submission written to ./working1/submission.csv")
```

The Coordinator submits the scripts generated by Coding Agents and their workspace to the community. Then, it proceeds to the second iteration. The report generated by the Analyzer for the second Coding Agent is:

```
Pipeline:
Our final pipeline is an Advanced Wavelet U-Net with CBAM attention modules, Composite Loss,
     and sliding-window inference. Specifically:

1. Data Preparation
   - Read all 112 noisy/clean train pairs, split 90/10 train/val by file indices (random seed
       42).
   - On-the-fly extract 25 random 256*256 patches per train image each epoch, with random
       horizontal/vertical flips and 90 degree rotations.
   - Validation uses a single full-image sliding-window pass (256 patches with 50% overlap) on
       the 10% hold-out.

2. Model Architecture
   - Haar DWT and inverse DWT implemented as fixed convolutional filters (LL, LH, HL, HH bands)
       .
   - Dual-branch encoder: separate CBAMResBlocks for the approximation (1 channel) and detail
       (3 channels) coefficients.
   - Two-level downsampling with MaxPool, interleaved CBAMResBlocks, and a 256-channel
       bottleneck.
   - Decoder with ConvTranspose upsampling and CBAMResBlock skip-connections (UpRes),
       recombining wavelet bands.
   - Final 1*1 convolution to 4 coefficient channels and inverse DWT back to image domain.

3. Composite Loss
   - L1(pixels) weighted by alpha=0.769634
   - L2(wavelet coefficients) weighted by beta=0.430968
   - L1(edge Sobel) weighted by gamma=0.250750

4. Training
   - Optimizer: AdamW(lr=1e-3, weight_decay=1e-4).
   - Scheduler: CosineAnnealingLR(T_max=30).
   - Batch size 8, 30 epochs, early stopping (patience=5) based on full-image RMSE on the val
       set.

5. Inference
   - Sliding-window full-image inference with 256*256 patches, stride=128, averaging overlapped
       pixels.
   - Generate submission.csv with id=image_row_col and float values.

6. Hyperparameter tuning
   - Optuna over (alpha, beta, gamma) on a 20-train/5-val mini-split for 10 trials, each with 2
       epochs and 10 patches/image.
   - Best found: alpha=0.769634, beta=0.430968, gamma=0.250750.

Final validation RMSE on 10% hold-out: ~0.012685.

References: CBAM (Woo et al., ECCV 2018), Haar DWT.

Summary:
- Haar DWT:
 Novelty: 2
 Rationale: Uses standard 1-level Haar wavelet decomposition.
 Feasibility: 10
 Rationale: Straightforward fixed conv implementation in PyTorch.
 Effectiveness: 6
 Rationale: Explicit frequency separation aids denoising of high-frequency noise.
 Efficiency: 9
 Rationale: Lightweight fixed kernels on GPU.
 Confidence: 8
 Rationale: Verified on toy and full runs.

- CBAMResBlock:
 Novelty: 5
 Rationale: Incorporates both channel & spatial attention for context-aware features.
 Feasibility: 8
 Rationale: Moderate complexity but supported by existing PyTorch modules.
 Effectiveness: 7
 Rationale: Improves focus on text strokes, empirically lowers RMSE.
 Efficiency: 7
 Rationale: Adds overhead (~1.5*), but fits within GPU budget.
 Confidence: 9
 Rationale: Confirmed across toy and full training.

- UpRes (ConvTranspose + CBAM):
 Novelty: 3
 Rationale: Standard upsampling with skip-connections and attention.
 Feasibility: 9
 Rationale: Simple extension of ResBlock.
```

```
  Effectiveness: 6
  Rationale: Maintains detail in reconstruction.
  Efficiency: 8
  Rationale: Reasonable GPU use.
  Confidence: 8
  Rationale: Working in end-to-end script.

- Wavelet UNet overall:
  Novelty: 7
  Rationale: Dual-branch U-Net in wavelet domain combining DWT, CBAM, skip connections.
  Feasibility: 7
  Rationale: Moderate engineering but fully implemented.
  Effectiveness: 8
  Rationale: Achieved RMSE ~0.0127, strong denoising.
  Efficiency: 7
  Rationale: 25 patches*8 batch; 30 epochs in ~20min.
  Confidence: 8
  Rationale: Reproducible on A6000.

- Composite Loss:
  Novelty: 6
  Rationale: Balances pixel, frequency, and edge domains.
  Feasibility: 8
  Rationale: Simple weighted sum.
  Effectiveness: 8
  Rationale: Lower RMSE than L1 or L2 alone.
  Efficiency: 7
  Rationale: Small overhead for extra transforms.
  Confidence: 9
  Rationale: Verified in tuning and full runs.

- Data Augmentation:
  Novelty: 3
  Rationale: Flips and rotations.
  Feasibility: 10
  Rationale: Trivial to implement.
  Effectiveness: 5
  Rationale: Standard but helpful for invariance.
  Efficiency: 9
  Rationale: CPU-level only.
  Confidence: 9
  Rationale: Common practice.

- CosineAnnealingLR & EarlyStopping:
  Novelty: 4
  Rationale: Standard learning-rate decay and val-based stop.
  Feasibility: 10
  Rationale: Built-in PyTorch.
  Effectiveness: 6
  Rationale: Helps convergence and prevents overfit.
  Efficiency: 9
  Rationale: Minimal overhead.
  Confidence: 9
  Rationale: Observed smoothed training curves.

- Sliding-window Inference:
  Novelty: 3
  Rationale: Ensures full-image coverage.
  Feasibility: 10
  Rationale: Straightforward loops.
  Effectiveness: 7
  Rationale: Accurate RMSE measurement, smooth outputs.
  Efficiency: 8
  Rationale: Reasonable inference time (~10 min).
  Confidence: 9
  Rationale: Verified end-to-end.

- Hyperparameter Tuning (Optuna):
  Novelty: 4
  Rationale: Automated search of loss weights.
  Feasibility: 8
  Rationale: Low-cost mini-training.
  Effectiveness: 7
  Rationale: Found better weights vs. defaults.
  Efficiency: 5
  Rationale: 10 trials * 2 epochs ~8 min.
  Confidence: 7
  Rationale: Verified chosen parameters.

Weaknesses and Suggestions:
```

1. EMA & TTA Bug: The EMA + 4-way TTA implementation caused RMSE blow-ups, likely due to incorrect accumulation. Before reintroducing, validate TTA on toy splits or isolate EMA without TTA.
2. Multi-Level Wavelets: Extend to 2-3 levels of DWT to capture coarser scales, with separate branches per level.
3. Advanced Loss Terms: Incorporate perceptual (VGG) or structural (SSIM) losses to encourage visual fidelity.
4. Dynamic Loss Weights: Schedule (alpha,beta,gamma) during training (e.g., start with high L1 , gradually emphasize wavelet).
5. Mixed Precision & Gradient Checkpointing: Further reduce memory and speed up with AMP and checkpointing deep layers.
6. Deeper/Ubiquitous Attention: Explore transformer-style attention (Swin, ViT) in encoder for global context.
7. Ensemble Models: Train multiple random seeds or architectures (ResUNet, UNet++) and average outputs.
8. Multi-Scale Inference: Use patches at multiple scales or whole-image passes via down/ upscaling for context.
9. Adaptive Overlap: Use weights at edges of patches to smoothly blend overlaps (Gaussian windows).
10. Distributed Hyperparameter Search: Scale Optuna across more trials or use BOHB for architecture search.
Implementing these will likely push RMSE even lower and improve robustness across document noise patterns.
Full code: ...(omitted)...

Following the brainstorming and idea refinement phases, the current idea memory is presented below. Notably, entries (3), (4), (5), and (7) are newly proposed by the agent, while the remaining ideas have been refined from those in the previous iteration.

(0) Patch-based convolutional super-resolution denoiser: a sliding-window, patch-based model that predicts multiple denoised and super-resolved pixels per patch. Architecture inspired by Waifu2x with progressively increasing filter counts, LeakyReLU activations, and He initialization for robust convergence.
(1) Advanced wavelet-domain dual-branch U-Net with CBAM attention and composite loss: decompose inputs via fixed 1-level DWT (LL, LH, HL, HH bands), encode approximation and detail separately with CBAM ResBlocks, decode and fuse via 1*1 convolutions, and train end-to-end using a weighted sum of pixel L1, wavelet-band L2, and edge L1 losses. Optimized with AdamW and cosine-annealing LR scheduling.
(2) GAN-based restoration framework: a ResNet-based generator and 70*70 PatchGAN discriminator trained with combined losses-L1 pixel loss, adversarial loss, stroke-consistency loss ( via frozen stroke-feature CNN), and perceptual OCR-feature loss. Includes R1 gradient penalty and spectral normalization for stability.
(3) Masked autoencoder with vision transformer for denoising: patchify each image into non-overlapping square tokens, randomly mask a high percentage, pretrain a ViT encoder (12 layers, hidden 768, 12 heads) plus light transformer decoder on L2 reconstruction of dirty images, then append an MLP head and fine-tune end-to-end on noisy->clean pairs with L1 pixel + differentiable OCR-confidence loss. Employ random block dropout and color jitter during fine-tuning; at inference use full-image encoding or averaged mask schedules.
(4) Conditional diffusion-based restoration: define a forward Gaussian-noise diffusion schedule, train a 5-level U-Net conditioned on the dirty image via channel concatenation and FiLM/cross-attention of sinusoidal timestep embeddings. Use the standard DDPM MSE loss with classifier-free guidance, and sample with a deterministic DDIM sampler (~50 steps). Optionally post-process with bilateral or median filtering to remove speckles.
(5) Learnable spectral gating in the Fourier domain: compute the 2D FFT of the dirty image, split its spectrum into low/mid/high radial bands, apply learnable complex masks per band , and modulate each by gate scalars predicted by a lightweight CNN on the dirty image. Recombine via inverse FFT and train end-to-end with L2 pixel loss plus a spectral-smoothness regularizer on the masks.
(6) Hypernetwork-modulated U-Net: extract per-image noise statistics (mean, std, skew, kurtosis, histogram bins), feed into an MLP hypernetwork that outputs FiLM scale (gamma) and shift (beta) parameters for selected convolutional feature maps of a base U-shaped CNN. Randomly augment noise levels during training; train end-to-end on noisy->clean with L1 loss and a small regularizer pushing gamma->1, beta->0. At inference compute stats per image, generate FiLM params, and denoise via the modulated U-Net.
(7) Blind-spot test-time adaptation: for each test image, randomly mask a subset of pixels and fine-tune a lightweight CNN with skip connections on the single image using masked L2 loss and augmentations for a few hundred gradient steps. Optionally ensemble multiple random initializations to reduce variance.
(8) Multi-model ensemble with diverse preprocessing: cluster trained models by their error patterns and combine their outputs. Apply different preprocessing pipelines (raw, background-removed, edge maps, thresholded) to the input, denoise with clustered sub-ensembles, and fuse predictions for robustness across noise characteristics.
(9) Enhanced augmentation and target refinement: simulate realistic scanning artifacts by applying 3D text deformations, lighting variations, and shadows to clean images. Identify and compensate for systematic artifacts in the provided 'clean' targets (e.g., single-pixel halos) by either modeling them as noise or adjusting ground-truth labels during training.

And solution drafts generated in this iteration are:

Pipeline 1: ResNet-34 Encoder U-Net with Multi-Scale Edge & Total-Variation Loss

Overview:
A robust baseline using a pretrained ResNet-34 backbone as a U-Net encoder fused with a light-weight decoder. Combines L1 loss, Sobel edge loss at multiple scales, and a total-variation regularizer to preserve text strokes while smoothing background noise. Mixed precision training and sliding-window inference ensure the entire pipeline runs in ~45 min on an A6000.

1. Data Preparation
 - Read all train noisy/clean PNGs, normalize to [0,1].
 - Extract on-the-fly 256*256 patches: random crop + random horizontal/vertical flips + 90 degree rotations.
 - 90/10 split by file indices (seed=42). Use batch size 8-16.

2. Model Architecture
 - Encoder: torchvision.models.resnet34(pretrained=True), first conv modified to 1->64 channels.
 - Decoder: four upsampling stages (ConvTranspose2d + Conv2d+BN+ReLU) mirroring ResNet blocks, with skip-connections from encoder layers.
 - Final conv 64->1 + Sigmoid.

3. Loss Function
 Let y_hat and y be predictions and targets.
 - L1Loss(y_hat,y)
 - Edge loss: L1 between Sobel(y_hat) and Sobel(y) at both full resolution and half resolution (downsample by 2).
 - TV: lambda*TV(y_hat) where TV = mean($|\nabla xy\_hat|+|\nabla yy\_hat|$).
 Total loss = alpha*L1 + beta*Edge_full + gamma*Edge_half + delta*TV, e.g. alpha=1.0, beta=0.5, gamma=0.25, delta=1e-5.

4. Optimization
 - Optimizer: AdamW(lr=1e-3, weight_decay=1e-4).
 - Scheduler: CosineAnnealingLR(T_max=25).
 - Mixed precision via torch.cuda.amp.
 - Early stopping on validation RMSE (patience=5).

5. Inference & Submission
 - Perform sliding-window inference on each test image with 256*256 patches, stride=128.
 - Average overlapping patches.
 - Clamp outputs to [0,1], write submission.csv with id=image_row_col.

Compute budget: ~20 min train + ~5 min inference.

Pipeline 2: Laplacian-Pyramid Multi-Scale Residual U-Net with Pyramid Loss

Overview:
A novel pyramid-domain network that decomposes images into multi-scale Laplacian bands, denoises each band via shared-weight residual blocks, and merges them back. Multi-level L1 losses focus the model on both coarse structures and fine text details. Efficient and fully end-to-end in PyTorch, training finishes in ~50 min on an A6000.

1. Data & Augmentation
 - Same data split and patch sampling (256*256, flips, rotations).

2. Laplacian Pyramid Transform (LPT)
 - On the fly, for each patch: create 2-level Gaussian pyramid using avg-pooling (scale 1->0.5->0.25), then compute Laplacian bands L0=(I-upsample(I/2)), L1=(I/2-upsample(I/4)), and a residual low band L2=I/4.
 - Stack [L0,L1,L2] as 3 input channels.

3. Network
 - Encoder-decoder U-Net with 3-channel input and 3-channel output, 4 down/upsampling stages, 64->128->256->512->256->128->64 channel widths.
 - Each block: two 3*3 Conv+BN+ReLU and a residual skip connection.

4. Reconstruction
 - Output is 3 residual corrections R0,R1,R2 to each band.
 - Reconstruct clean patch:
  a0 = L0 + R0; a1 = L1 + R1; a2 = L2 + R2;
  I = a0 + up(a1) + up(up(a2)).

5. Loss Function
 - Per-band L1 losses: w0*L1(a0,H0) + w1*L1(a1,H1) + w2*L1(a2,H2), where Hk are ground-truth bands and w0=1.0, w1=0.5, w2=0.25.
 - Pixel L1: lambda_pix*L1(I, Igt) with lambda_pix=0.1.

6. Training
 - AdamW(lr=5e-4, weight_decay=1e-5), one-cycle scheduler (max_lr=5e-4, epochs=30).
 - Batch size 8, mixed precision.
 - Early stopping on val RMSE (patience=6).

```
7. Inference
 - On full test images, run sliding-window with 256*256 patches, stride=128.
 - For each patch, apply pyramid -> model -> reconstruct -> accumulate and average.
 - Clamp to [0,1], write submission.csv.

Compute budget: ˜25 min training + ˜8 min inference.
```

The best code proposed in the second iteration is:

```python
import os, random, csv
import numpy as np
from glob import glob
from PIL import Image

import torch
import torch.nn as nn
import torch.nn.functional as F
from torch.utils.data import Dataset, DataLoader
from torchvision import transforms, models as tv_models
from torch.cuda.amp import autocast, GradScaler

# 1) Reproducibility
SEED = 42
random.seed(SEED)
np.random.seed(SEED)
torch.manual_seed(SEED)
torch.cuda.manual_seed_all(SEED)

DEVICE = torch.device("cuda" if torch.cuda.is_available() else "cpu")

# 2) Paths
TRAIN_NOISY = "./input/train"
TRAIN_CLEAN = "./input/train_cleaned"
TEST_DIR = "./input/test"
WORK_DIR = "./working0"
os.makedirs(WORK_DIR, exist_ok=True)

# 3) Hyperparameters
PATCH_SIZE = 256
STRIDE = 128
BATCH_SIZE = 12
LR, WD = 1e-3, 1e-4
MAX_EPOCHS = 50
PATIENCE = 10
T_MAX = 50 # for LR scheduler

# 4) Loss-weight constants (from tuning)
w1, w2, w3, w4 = 1.0, 0.5, 0.25, 1e-5
lambda_aux = 0.4394633936788146
lambda_mse = 0.1312037280884873
lambda_ssim = 0.031198904067240532
lambda_ssim2 = lambda_ssim / 2

# 5) Dataset + augmentations
class OCRDataset(Dataset):
    def __init__(self, noisy_list, clean_list, ps, train):
        self.noisy, self.clean = noisy_list, clean_list
        self.ps, self.train = ps, train
        self.to_tensor = transforms.ToTensor()
        self.aug = transforms.Compose(
            [
                transforms.RandomChoice(
                    [
                        transforms.RandomHorizontalFlip(1.0),
```

```python
                    transforms.RandomVerticalFlip(1.0),
                    transforms.RandomRotation(90),
                    transforms.RandomRotation(180),
                    transforms.RandomRotation(270),
                ]
            ),
            transforms.RandomApply([transforms.GaussianBlur(3, (0.1, 2.0)
                )], p=0.3),
            transforms.RandomApply([transforms.RandomAdjustSharpness(2.0)
                ], p=0.3),
        ]
    )

    def __len__(self):
        return len(self.noisy)

    def __getitem__(self, i):
        n = Image.open(self.noisy[i]).convert("L")
        c = Image.open(self.clean[i]).convert("L")
        w, h = n.size
        # pad
        if w < self.ps or h < self.ps:
            pad = (0, 0, max(0, self.ps - w), max(0, self.ps - h))
            n = transforms.functional.pad(n, pad, fill=255)
            c = transforms.functional.pad(c, pad, fill=255)
            w, h = n.size
        # crop
        if self.train:
            x = random.randint(0, w - self.ps)
            y = random.randint(0, h - self.ps)
        else:
            x = (w - self.ps) // 2
            y = (h - self.ps) // 2
        n = n.crop((x, y, x + self.ps, y + self.ps))
        c = c.crop((x, y, x + self.ps, y + self.ps))
        if self.train and random.random() < 0.5:
            n = self.aug(n)
            c = self.aug(c)
        return self.to_tensor(n), self.to_tensor(c)

# 6) Prepare train/val split
noisy_files = sorted(glob(f"{TRAIN_NOISY}/*.png"))
clean_files = [f"{TRAIN_CLEAN}/" + os.path.basename(x) for x in
    noisy_files]
N = len(noisy_files)
idx = list(range(N))
random.shuffle(idx)
ntr = int(0.9 * N)
tr_idx, va_idx = idx[:ntr], idx[ntr:]
train_noisy = [noisy_files[i] for i in tr_idx]
train_clean = [clean_files[i] for i in tr_idx]
val_noisy = [noisy_files[i] for i in va_idx]
val_clean = [clean_files[i] for i in va_idx]

train_ds = OCRDataset(train_noisy, train_clean, PATCH_SIZE, train=True)
val_ds = OCRDataset(val_noisy, val_clean, PATCH_SIZE, train=False)
train_loader = DataLoader(
    train_ds, batch_size=BATCH_SIZE, shuffle=True, num_workers=4,
        pin_memory=True
)
val_loader = DataLoader(
    val_ds, batch_size=BATCH_SIZE, shuffle=False, num_workers=4,
        pin_memory=True
)
```

```
117
118   # 7) Sobel, TV, SSIM helpers
119   sob_x = (
120       torch.tensor([[1, 0, -1], [2, 0, -2], [1, 0, -1]], dtype=torch.float32
              )
121       .view(1, 1, 3, 3)
122       .to(DEVICE)
123   )
124   sob_y = sob_x.transpose(2, 3)
125
126
127   def sobel(x):
128       gx = F.conv2d(x, sob_x, padding=1)
129       gy = F.conv2d(x, sob_y, padding=1)
130       return torch.sqrt(gx * gx + gy * gy + 1e-6)
131
132
133   def total_variation(x):
134       dh = (x[:, :, 1:, :] - x[:, :, :-1, :]).abs().mean()
135       dw = (x[:, :, :, 1:] - x[:, :, :, :-1]).abs().mean()
136       return dh + dw
137
138
139   def ssim_map(a, b, C1=0.01**2, C2=0.03**2):
140       mu_a = F.avg_pool2d(a, 3, 1, 1)
141       mu_b = F.avg_pool2d(b, 3, 1, 1)
142       sa = F.avg_pool2d(a * a, 3, 1, 1) - mu_a * mu_a
143       sb = F.avg_pool2d(b * b, 3, 1, 1) - mu_b * mu_b
144       sab = F.avg_pool2d(a * b, 3, 1, 1) - mu_a * mu_b
145       num = (2 * mu_a * mu_b + C1) * (2 * sab + C2)
146       den = (mu_a * mu_a + mu_b * mu_b + C1) * (sa + sb + C2)
147       return num / (den + 1e-8)
148
149
150   def ssim_loss(a, b):
151       return 1.0 - ssim_map(a, b).mean()
152
153
154   # 8) loss_terms
155   l1_loss = nn.L1Loss()
156   mse_loss = nn.MSELoss()
157
158
159   def loss_terms(pred, target):
160       L1v = l1_loss(pred, target)
161       MSEv = mse_loss(pred, target)
162       Ef = l1_loss(sobel(pred), sobel(target))
163       p2, t2 = F.avg_pool2d(pred, 2), F.avg_pool2d(target, 2)
164       Eh = l1_loss(sobel(p2), sobel(t2))
165       TVv = total_variation(pred)
166       return L1v, MSEv, Ef, Eh, TVv
167
168
169   # 9) Model w/ deep supervision
170   class ResUNetDS(nn.Module):
171       def __init__(self):
172           super().__init__()
173           r34 = tv_models.resnet34(pretrained=True)
174           self.enc0 = nn.Conv2d(1, 64, 7, 2, 3, bias=False)
175           self.enc0.weight.data = r34.conv1.weight.data.mean(dim=1, keepdim=
                  True)
176           self.bn0, self.relu0, self.pool0 = r34.bn1, r34.relu, r34.maxpool
177           self.enc1, self.enc2 = r34.layer1, r34.layer2
178           self.enc3, self.enc4 = r34.layer3, r34.layer4
179
```

```python
        def up(i, o):
            return nn.ConvTranspose2d(i, o, 2, 2)

        def cb(i, o):
            return nn.Sequential(
                nn.Conv2d(i, o, 3, 1, 1, bias=False),
                nn.BatchNorm2d(o),
                nn.ReLU(inplace=True),
                nn.Conv2d(o, o, 3, 1, 1, bias=False),
                nn.BatchNorm2d(o),
                nn.ReLU(inplace=True),
            )

        self.up4, self.dec4 = up(512, 256), cb(256 + 256, 256)
        self.up3, self.dec3 = up(256, 128), cb(128 + 128, 128)
        self.up2, self.dec2 = up(128, 64), cb(64 + 64, 64)
        self.aux_up, self.aux_out = up(64, 64), nn.Conv2d(64, 1, 1)
        self.up1, self.dec1 = up(64, 64), cb(64 + 64, 64)
        self.up0, self.outc = up(64, 64), nn.Conv2d(64, 1, 1)
        self.sig = nn.Sigmoid()

    def forward(self, x):
        x0 = self.relu0(self.bn0(self.enc0(x)))
        x1 = self.pool0(x0)
        x2 = self.enc1(x1)
        x3 = self.enc2(x2)
        x4 = self.enc3(x3)
        x5 = self.enc4(x4)

        d4 = self.dec4(torch.cat([self.up4(x5), x4], dim=1))
        d3 = self.dec3(torch.cat([self.up3(d4), x3], dim=1))
        d2 = self.dec2(torch.cat([self.up2(d3), x2], dim=1))
        aux = self.sig(self.aux_out(self.aux_up(d2)))
        d1 = self.dec1(torch.cat([self.up1(d2), x0], dim=1))
        main = self.sig(self.outc(self.up0(d1)))
        return main, aux

model = ResUNetDS().to(DEVICE)

# 10) Optimizer, scheduler, scaler
optimizer = torch.optim.AdamW(model.parameters(), lr=LR, weight_decay=WD)
scheduler = torch.optim.lr_scheduler.CosineAnnealingLR(optimizer, T_max=
    T_MAX)
scaler = GradScaler()

# 11) Training + snapshot saving
best_rmse = float("inf")
patience = 0
snap_epochs = set([10, 20, 30, 40, 50])

for epoch in range(1, MAX_EPOCHS + 1):
    model.train()
    train_loss = 0.0
    for noisy_img, clean_img in train_loader:
        noisy_img, clean_img = noisy_img.to(DEVICE), clean_img.to(DEVICE)
        optimizer.zero_grad()
        with autocast():
            main_pred, aux_pred = model(noisy_img)
            L1v, MSEv, Ef, Eh, TVv = loss_terms(main_pred, clean_img)
            s1 = ssim_loss(main_pred, clean_img)
            p2, t2 = F.avg_pool2d(main_pred, 2), F.avg_pool2d(clean_img, 2)
            s2 = ssim_loss(p2, t2)
            main_loss = (
                w1 * L1v
```

```python
                    + lambda_mse * MSEv
                    + w2 * Ef
                    + w3 * Eh
                    + w4 * TVv
                    + lambda_ssim * s1
                    + lambda_ssim2 * s2
                )
                aux_up = F.interpolate(
                    aux_pred,
                    size=clean_img.shape[-2:],
                    mode="bilinear",
                    align_corners=False,
                )
                La, Ma, Ea, Eh2, TVa = loss_terms(aux_up, clean_img)
                sa = ssim_loss(aux_up, clean_img)
                pa, ca = F.avg_pool2d(aux_up, 2), F.avg_pool2d(clean_img, 2)
                sa2 = ssim_loss(pa, ca)
                aux_loss = (
                    w1 * La
                    + lambda_mse * Ma
                    + w2 * Ea
                    + w3 * Eh2
                    + w4 * TVa
                    + lambda_ssim * sa
                    + lambda_ssim2 * sa2
                )
                loss = main_loss + lambda_aux * aux_loss
            scaler.scale(loss).backward()
            scaler.step(optimizer)
            scaler.update()
            train_loss += loss.item()
        scheduler.step()

        # validation
        model.eval()
        se, count = 0.0, 0
        with torch.no_grad():
            for noisy_img, clean_img in val_loader:
                noisy_img, clean_img = noisy_img.to(DEVICE), clean_img.to(DEVICE
                    )
                with autocast():
                    pred, _ = model(noisy_img)
                se += ((pred - clean_img) ** 2).sum().item()
                count += pred.numel()
        val_rmse = np.sqrt(se / count)
        print(
            f"Epoch {epoch}: TrainLoss={train_loss/len(train_loader):.4f}, 
                ValRMSE={val_rmse:.6f}"
        )

        # best + snapshot
        if val_rmse < best_rmse:
            best_rmse = val_rmse
            torch.save(model.state_dict(), os.path.join(WORK_DIR, "best_full.
                pth"))
            patience = 0
        else:
            patience += 1
        if epoch in snap_epochs:
            torch.save(model.state_dict(), os.path.join(WORK_DIR, f"snap_{epoch
                }.pth"))
        if patience >= PATIENCE:
            print("Early stopping.")
            break
```

```python
305  print("Best validation RMSE:", best_rmse)
306
307  # 12) Ensemble load
308  ckpts = ["best_full.pth"] + sorted(
309      [f for f in os.listdir(WORK_DIR) if f.startswith("snap_")],
310      key=lambda x: int(x.split("_")[1].split(".")[0]),
311  )[-2:]
312  ensemble_nets = []
313  for ck in ckpts:
314      net = ResUNetDS().to(DEVICE)
315      net.load_state_dict(torch.load(os.path.join(WORK_DIR, ck)))
316      net.eval()
317      ensemble_nets.append(net)
318
319
320  # 13) Sliding-window ensemble inference
321  def ensemble_infer(img_arr):
322      h, w = img_arr.shape
323      inp = torch.from_numpy(img_arr / 255.0).unsqueeze(0).unsqueeze(0).to(
324          DEVICE)
325      ph = (PATCH_SIZE - h % STRIDE) % STRIDE
326      pw = (PATCH_SIZE - w % STRIDE) % STRIDE
327      inp = F.pad(inp, (0, pw, 0, ph), mode="reflect")
328      _, _, H, W = inp.shape
329      out = torch.zeros_like(inp)
330      wt = torch.zeros_like(inp)
331      for y in range(0, H - PATCH_SIZE + 1, STRIDE):
332          for x in range(0, W - PATCH_SIZE + 1, STRIDE):
333              patch = inp[:, :, y : y + PATCH_SIZE, x : x + PATCH_SIZE]
334              preds = []
335              with torch.no_grad(), autocast():
336                  for net in ensemble_nets:
337                      p, _ = net(patch)
338                      preds.append(p)
339              avg_p = torch.stack(preds, 0).mean(0)
340              out[:, :, y : y + PATCH_SIZE, x : x + PATCH_SIZE] += avg_p
341              wt[:, :, y : y + PATCH_SIZE, x : x + PATCH_SIZE] += 1.0
342      out = out / wt
343      out = out[:, :, :h, :w]
344      return out.detach().cpu().numpy().squeeze()
345
346
347  # 14) Write submission.csv
348  submission_path = os.path.join(WORK_DIR, "submission.csv")
349  with open(submission_path, "w", newline="") as f:
350      writer = csv.writer(f)
351      writer.writerow(["id", "value"])
352      for tf in sorted(
353          glob(f"{TEST_DIR}/*.png"), key=lambda x: int(os.path.basename(x).
354              split(".")[0])
355      ):
356          img_id = os.path.basename(tf).split(".")[0]
357          img = np.array(Image.open(tf).convert("L"), dtype=np.float32)
358          den = ensemble_infer(img)
359          H, W = den.shape
360          for i in range(H):
361              for j in range(W):
362                  writer.writerow([f"{img_id}_{i+1}_{j+1}", f"{den[i,j]:.6f}"])
363  print("Submission saved to", submission_path)
```