# OpenReview forum: "CoMind: Towards Community-Driven Agents for Machine Learning Engineering"
_ICLR.cc/2026/Conference — ICLR 2026 Poster_

### Official Review · Reviewer_4mfQ · 2025-10-24

**Soundness:** 3
**Presentation:** 2
**Contribution:** 3
**Rating:** 6
**Confidence:** 4

**Summary:**

The paper proposes CoMind, a multi-agent LLM system for ML engineering that integrates external community knowledge. It also proposes MLE-Live, a framework simulating Kaggle-style community environments with public code and discussions. CoMind uses five specialized agents (Coordinator, Analyzer, Idea Proposer, Coding Agent, Evaluator) to iteratively generate, refine, and evaluate ML solutions. The authors evaluate it on 75 past Kaggle competitions (MLE-Bench) and 8 live competitions. It achieves a 36% medal rate, better than some other open-source baselines (R&D-Agent, ML-Master, AIDE). In live tests, CoMind ranks in the top 10% on average, outperforming the majority of human competitors. Ablation shows strong dependence on public resources, removal causes major drops in submission success and performance. It generates longer, more complex code than baselines, which could mean a focus on exploration but potentially lower efficiency.

**Strengths:**

The paper's strenghts include the MLE-Live setup with realistic access to public resources. They show strong empirical results on both retrospective and live tasks, with a clear system design, their multi-agent division of roles aligns with human research workflows. The results are there, demonstrating practical success in real Kaggle leaderboards.

The results overall on Kaggle are strong enough that I am leaning towards accept, but I still feel like their approach could be overfitting to some Kaggle property and urgently needs more validation to show its generalisability on another, similar type of challenge (see weaknesses)

**Weaknesses:**

The weaknesses include the efficiency tradeoff with CoMind producing longer and more complex code, with higher compute/time cost.
I would have liked to see a more thorough analysis with limited tokens, and of runtime efficiency. As such, the scaleability is not really clear to me. It may be too heavily dependent of Kaggle-specific properties, it would have been good to include (even if in a limited manner) some other, similar use case as well, simply to avoid overfitting on some weird Kaggle idiosyncracy. Some of the improvements over baselines are rather modest given the additional architectural complexity.

**Questions:**

Can you provide more information on the runtime and computational costs per competition (tokens, wall-clock, GPU hours)?

Does CoMind's higher code complexity correlate with actual performance gains or just verbosity?

How would the framework generalize beyond Kaggle-style settings (e.g., real ML research or open-ended tasks)?

---

> ### Author Response · Authors · 2025-11-21
>
> We appreciate the acknowledgement of CoMind's strong empirical performance and realistic multi-agent design. For your questions:
>
> > Q1: Can you provide more information on the runtime and computational costs per competition (tokens, wall-clock, GPU hours)?
>
> A1: We thank the reviewer for raising this point. Following the settings of MLE-Bench, each competition is executed under a fixed resource budget of 32 vCPUs, 1× A6000 GPU, and a 24-hour wall-clock limit (with each code-execution session capped at 5 hours). Across the 75 MLE-Bench tasks, a single run of CoMind on one competition incurs an average API cost of **$32.25 ± 19.43**. We include a detailed breakdown of token usage in **Appendix C** for completeness and transparency.
>
> > Q2: Does CoMind's higher code complexity correlate with actual performance gains or just verbosity?
>
> A2: Yes, our results show that CoMind's higher code complexity corresponds to meaningful performance gains rather than verbosity.
> As shown in Figure 4, CoMind's more complex pipelines consistently achieve higher Win Rate, Any Medal, and Above Median scores than baseline agents under the same backbone and resource constraints.
> In addition, Appendix A provides objective complexity metrics (Cyclomatic Complexity, Halstead Difficulty/Effort, Pylint score), which collectively demonstrate that CoMind's code reflects richer model design, feature engineering, and training logic rather than unnecessary expansion.
>
>
> > Q3: How would the framework generalize beyond Kaggle-style settings (e.g., real ML research or open-ended tasks)?
>
> A3: Besides the simulated community, we deliberately kept other parts of CoMind simple. The core components of the simulated community (time-stamped artifacts, resource governance, community simulation, evaluator abstraction) can be extended naturally beyond Kaggle. In real ML research or open-ended domains, the same mechanisms can be instantiated using other forms of public knowledge, e.g., GitHub issues and arXiv discussions. The Coordinator/Analyzer/Idea Proposer workflow is model-agnostic and does not rely on Kaggle-specific structures; the only Kaggle-specific component is the grader, which can be replaced by any domain-appropriate evaluation function.

---

> > ### Comment · Reviewer_4mfQ · 2025-11-25
> >
> > Thank you for your response. Regarding your A3, let me just add that even though mechanisms often seem domain-agnostic and easily transferable, in my experience that often does not turn out to be the case. Some mechanisms are, inadvertently, overfitted to specific environments, which may include Kaggle.

---

> > > ### Author Response · Authors · 2025-12-03
> > >
> > > To directly assess CoMind’s extensibility beyond Kaggle-style environments, we conducted an additional experiment on **CoBench** [1], a dataset of 36 combinatorial optimization problems that is structurally different from Kaggle competitions and does not involve notebooks, discussions, or leaderboard dynamics. Using gpt-5-mini as a shared backbone, we compared CoMind (without access to public resources) against **AIDE** [2], **FunSearch** [3], and **Greedy Refinement** (iteratively prompting the LLM to refine the current best solution). Following the settings of CoBench, each agent operates through 64 research steps, receiving feedbacks on a test set at each step and has a 10-second execution time limitation. Across 9 representative CoBench tasks, CoMind's normalized score achieves competitive performance and, on average, outperforms the baselines, as shown below.
> > >
> > > | **Task**                       | **CoMind** | **AIDE**  | **FunSearch** | **Greedy Refinement** |
> > > | ------------------------------ | ---------- | --------- | ------------- | --------------------- |
> > > | Crew scheduling                | **0.915**  | 0.448     | 0.546         | 0.602                 |
> > > | Graph colouring                | 0.879      | 0.850     | 0.893         | **0.968**             |
> > > | Constrained guillotine cutting | 0.975      | 0.911     | **0.993**     | 0.989                 |
> > > | MIS                            | 0.909      | **0.932** | 0.860         | 0.874                 |
> > > | Aircraft landing               | **0.865**  | 0.863     | 0.760         | 0.378                 |
> > > | Bin packing (1D)               | 0.903      | 0.925     | **0.975**     | 0.821                 |
> > > | Euclidean Steiner problem      | 0.690      | 0.636     | 0.701         | **0.760**             |
> > > | Set covering                   | **0.922**  | 0.887     | 0.918         | 0.916                 |
> > > | TSP                            | **0.923**  | 0.606     | 0.860         | 0.832                 |
> > > | **Avg**                        | **0.887**  | 0.784     | 0.834         | 0.793                 |
> > >
> > > These results provide concrete evidence that CoMind’s scaffold remains effective in domains different from Kaggle, thereby directly addressing the concern about possible overfitting. While this does not claim universal generalization, it demonstrates that the core architecture is not limited to Kaggle-like environments. We will include these additional experiments and clarifications in the final version.
> > >
> > > Reference:
> > >
> > > [1] Sun, Weiwei, et al. *Co-bench: Benchmarking language model agents in algorithm search for combinatorial optimization.* arXiv:2504.04310 (2025).
> > >
> > > [2] Jiang, Zhengyao, et al. *AIDE: AI-driven exploration in the space of code.* arXiv:2502.13138 (2025).
> > >
> > > [3] Romera-Paredes, Bernardino, et al. *Mathematical discoveries from program search with large language models.* *Nature* 625.7995 (2024): 468–475.

---

### Official Review · Reviewer_vka8 · 2025-10-31

**Soundness:** 4
**Presentation:** 4
**Contribution:** 3
**Rating:** 6
**Confidence:** 4

**Summary:**

The paper CoMind introduces MLE-Live, a benchmark designed to evaluate agents’ ability to leverage shared knowledge in realistic, community-driven Kaggle-style settings, and proposes a multi-agent system that simulates and contributes to such a research community. CoMind comprises five specialized agents—Coordinator, Analyzer, Idea Proposer, Coding Agent, and Evaluator—that iteratively synthesize ideas, implement pipelines, and refine solutions by integrating public discussions and code. Experiments on 75 Kaggle competitions from MLE-Bench show that CoMind achieves a 36% medal rate, surpassing prior systems like Neo and R&D-Agent, and in eight live competitions, it ranks within the top 7.35% on average, outperforming 92.6% of human competitors. The study demonstrates that community-augmented collaboration significantly enhances autonomous ML engineering performance.

**Strengths:**

- The paper introduces the novel concept of community-driven evaluation and agent collaboration in machine learning engineering. MLE-Live and CoMind simulate social learning and information exchange, a key feature of real-world research. I think this is quite meaningful.
- The paper is well-organized and readable.
- Its strong empirical gains and reproducible open-world setup could influence future benchmarks and frameworks in agentic AI, AutoML, and collaborative reasoning.
- I like the ablation. One downside of the main table is that the base model is all over the place. The ablation helps to put things in perpsective.

Overall I think the paper has its merit. A MLE benchmark that includes discussion, notebooks can be a useful asset for future agentic research.

**Weaknesses:**

```Gain came from the community data```

First I want to make the point that I believe a MLE-Bench with public resource is good. I have nothing against that. This weakness is mainly concern about the scaffolds proposed. In figure 4 there is the ablation where CoMind w R is much better than CoMind w/o R. CoMind w/o R is roughly equivalent to AIDE with the same resource and backend model. The improvement from scaffold itself is constrained. It would be great to see more evidences that CoMind is a better scaffold than others.

One evidence I think would be helpful is to actually use the same model (o4-mini) for the main table or at least for a couple top baselines in the main table. This is to eliminate the factors of backend llm. Another thing is to let other agents to have some kind of overall view on the public accessible resources. The current setup in the ablation study has a problem that top voted notebooks may not be the most helpful/effective solution. Having that ablation would greatly help the scaffold's case.

```More evidences of collaborative reasoning```

What reasoning modes do they workflow commonly show? Does it take the strongest notebook and improve upon it? Making those clear is crucial.

**Questions:**

- Is the benchmark going to be public for people to use soon?

---

> ### Author Response · Authors · 2025-11-21
>
> We appreciate the recognition of the novelty of community-driven evaluation and the strong empirical performance of CoMind. For your concerns and questions:
>
> > W1: First I want to make the point that I believe a MLE-Bench with public resource is good. I have nothing against that. This weakness is mainly concern about the scaffolds proposed. In figure 4 there is the ablation where CoMind w R is much better than CoMind w/o R. CoMind w/o R is roughly equivalent to AIDE with the same resource and backend model. The improvement from scaffold itself is constrained. It would be great to see more evidences that CoMind is a better scaffold than others.
> >
> > One evidence I think would be helpful is to actually use the same model (o4-mini) for the main table or at least for a couple top baselines in the main table. This is to eliminate the factors of backend llm. Another thing is to let other agents to have some kind of overall view on the public accessible resources. The current setup in the ablation study has a problem that top voted notebooks may not be the most helpful/effective solution. Having that ablation would greatly help the scaffold's case.
>
> A1:  We thank the reviewer for these thoughtful suggestions. We agree that isolating the effect of the scaffold from the benefit of external resources is an important question. Our design intentionally keeps the underlying agent simple (a ReAct-style agent) so that CoMind primarily differs from AIDE in how it exploits collective knowledge.
>
> We agree that using a shared backbone or enabling resource access for more baselines would be ideal. However, re-running the full MLE-Bench with all methods is quite expensive, and several top-performing agents are closed-source. For this reason, our ablation acts as a controlled comparison: it fixes the backbone and resource budget to directly measure scaffold-level differences between CoMind and AIDE.
>
> Finally, our simulated community mechanism is designed to leverage collective knowledge and is orthogonal to the scaffolds used by other agents. Extending this mechanism to more sophisticated scaffolds is valuable future work, and we have noted this explicitly in the limitations and future directions section.
>
> > W2: What reasoning modes do they workflow commonly show? Does it take the strongest notebook and improve upon it? Making those clear is crucial.
>
> A2: We provide a detailed case study in **Appendix I**. CoMind typically exhibits two reasoning modes: In the early stages, it often attempts to reproduce or lightly extend high-ranked public notebooks to establish a verified baseline. In the later stages, it synthesizes techniques from multiple notebooks together with novel ideas generated during brainstorming, selecting among them using the Evaluator’s metric.
>
> CoMind does not always build on the single strongest notebook since some public discussions or notebooks may be outdated or incorrect. The Evaluator ensures each candidate is judged by actual performance, so the final solution often diverges from any individual public notebook.
>
> > Q1: Is the benchmark going to be public for people to use soon?
>
> A3: Yes. We will open-source the full MLE-Live framework, including all tooling needed to construct live evaluation environments. We will also release the complete set of public discussions and code artifacts for the 75 competitions from MLE-Bench.

---

### Official Review · Reviewer_41Ga · 2025-11-01

**Soundness:** 3
**Presentation:** 2
**Contribution:** 3
**Rating:** 2
**Confidence:** 4

**Summary:**

The paper proposes MLE-Live, a live evaluation framework that can evaluate AI agents to gather knowledge from a simulated Kaggle research community. The paper then proposes CoMind, a multi-agent system that can combine external knowledge. The CoMind simulates a Kaggle community, including a coordinator, an analyzer, an idea proposer, and an evaluator. The paper compares against multiple SOTA agent frameworks. The paper also includes an ablation study, qualitative and quantitative analysis.

**Strengths:**

1. The paper provides a live evaluation framework, which contains not only competition, but also discussions and kernels. The paper cut the data before the competition deadline to mitigate data leakage.
2. The CoMind has an analyzer to distill knowledge from community artifacts. The paper also evaluates the eight Live competitions. Apart from quantitive analysis, the paper has some qualitative analysis on task category, winrate, and code complexity.
3. The paper provides the code and its model. In the appendix, the paper shows training details, additional evaluation details, and detailed case study.

**Weaknesses:**

1. The evaluation metrics follow the standard MLE-Bench for the main table. The ablation study only reports the win rate. I did not see any metrics directly related to the collaborative nature of agents. The novelty of MLE-Live seems to be limited.
2. Some parts of the paper are not clear. For example, what are the tasks included in the MLE-Live? The paper did not list any statistics or any details of those competitions. The paper also seems to categorize the tasks into three levels. However, it is unclear how those levels are determined, whether through the complexity of tasks or datasets. The external knowledge shared in the MLE-Live is also unclear. What is the average length of a discussion? What are their topics? The main architecture is very high-level and purely prompt-based. The paper also fail to provide cost analysis.
3. Compared to other benchmark papers, the evaluation is pretty limited. For example, in MLE0Dojo https://openreview.net/pdf?id=5W5mFU4oMO, they evaluate eight different backbones and provide task difficulty analysis for different domains. Using code length to evaluate code complexity instead of human annotation seems to be too coarse. The paper also fails to include any error analysis for the failure case.
4. The paper fails to include a use of LLMs section. The evaluated closed source LLMs are also not SOTA>

**Questions:**

See weakness

---

> ### Author Response · Authors · 2025-11-21
>
> We thank the reviewer for the detailed feedback and address each concern below.
>
> > W1: Novelty and collaborative evaluation metrics
>
> A1:  We appreciate the reviewer's feedback. We respectfully clarify that our goal is **not to update MLE-Bench** but to introduce a **distinct evaluation dimension** that existing benchmarks do not consider: **how agents leverage time-bounded, community-provided knowledge**.
>
> Novelty: Prior work evaluates agents strictly in **closed-world settings**, where access to external code or discussions is disallowed. In contrast, our work is the first framework to formalize Open-World ML Engineering. This novelty is two-fold: **(1) The Evaluation (MLE-Live):** MLE-Live is the first framework that provides agents with time-stamped public kernels and discussions, enabling controlled access to community artifacts without post-deadline leakage. **(2) The Agent (CoMind):** Unlike baselines that simply run independently, CoMind's community evolves as (i) the agent iterates and (ii) more kernels are published by human competitors. This dynamic, continuously updating community setting is what enables the study of "collaborative" behavior, an ability that existing autonomous ML agents do not possess.
>
> Regarding the measurement of collaboration, we posit that in competitive ML engineering, the measure of "collaborative success" is the performance uplift gained from external knowledge. While standard MLE-Bench metrics (Any Medal, etc.) are used in the main table for comparability, the ablation study directly reflects the collaborative dimension. We have evaluated metrics such as Win Rate, Any Medal, and Above Median to measure how effectively an agent converts community knowledge into performance gains. The dramatic drop in performance when removing external resources (CoMind w/o R), and the limited improvements of simpler resource-augmented baselines (AIDE+RAG, AIDE+Code), quantitatively demonstrate that collaboration with external artifacts is crucial, and CoMind's community-aware design is uniquely effective at exploiting such resources.
>
> > W2: Missing clarity on tasks, difficulty levels, external resources, and cost
>
> A2: We thank the reviewer for pointing out these clarity issues. We have substantially expanded the presentation in the revision:
> - Added **Appendix E** summarizing statistics of all 75 MLE-Bench competitions used within MLE-Live, including task domains and difficulties. We clarified that the task difficulty follows exactly the definitions used in the original MLE-Bench paper, which categorizes competitions based on human evaluation results. We have revised our paper to explicitly clarify this definition.
> - Added statistics of MLE-Live community artifacts in **Appendix F**, including number of kernels, discussions, average code length and votes distribution for each specific competition.
> - Provided details of **API cost**. A full run over one competition costs **$32.25 ± 19.43** in API usage, now explicitly reported in **Appendix C**. The hardware restrictions for CoMind are described in detail in the Implementation Setup in Sections 5.1 and 6.1.
>
> > W3: Evaluation scope, code complexity metrics, and lack of error analysis
>
> A3:  We address the three aspects below:
> 1. Scope Difference: MLE-Dojo is designed as a **dataset-centric benchmark**, focusing on comparing LLM backbones and characterizing task difficulty across domains. Our work has a different purpose: evaluating and designing agents that interact with **community-provided artifacts**. Because our research question concerns how agents use external resources, varying LLM backbones would obscure the effect of the collaborative mechanisms. Thus, we fix the backbone to isolate the contribution from community-aware design.
> 2. Code Complexity Metric: We agree that code length alone is insufficient. However, we respectfully clarify that we did not rely solely on length. As detailed in **Appendix B**, we utilized a suite of objective, industry-standard software engineering metrics, including Cyclomatic Complexity, Pylint score, and Halstead Volume/Difficulty/Effort. We prioritized these over human annotation to ensure reproducibility and avoid subjective bias. Furthermore, Figure 4 demonstrates a strong correlation between these complexity metrics and win rates, confirming that they successfully capture meaningful engineering sophistication rather than trivial verbosity.
> 3. Error Analysis: We now include a dedicated error analysis in **Appendix G**, categorizing major failure modes and describing common issues observed across tasks
>
> > W4: The paper fails to include a use of LLMs section. The evaluated closed source LLMs are also not SOTA
>
> A4: We have added a Use of LLMs section. Our primary contributions, the MLE-Live evaluation framework and the CoMind architecture, are orthogonal to backbone LLM choice. The empirical gains reflect the benefits of community-aware design, rather than reliance on the latest proprietary SOTA LLMs.

---

> > ### Comment · Reviewer_41Ga · 2025-11-26
> >
> > Thanks for your detailed explanation! I decide to raise my score to 6.

---

### Official Review · Reviewer_d6GG · 2025-11-03

**Soundness:** 2
**Presentation:** 2
**Contribution:** 2
**Rating:** 6
**Confidence:** 3

**Summary:**

This work tackles the problem of automating machine learning engineering. Differently from most existing work, the authors take a community-driven approach and setup where AI agents can perceive real-time information and knowledge from external (human) communities. To this end, they first present their own framework, named MLE-Live. Building on top of MLE-Bench, MLE-Live simulates a live stream of publicly available information, including source codes and discussions (as for Kaggle competitions), namely "community artifacts," which are based on the pieces of information that were uploaded before the competition deadlines. In addition, the authors propose CoMind, a multi-agent framework that leverages such human inputs for machine learning engineering tasks and is made of five sub-agents with different roles (Coordinator, Analyzer, Idea Proposer, Coding Agents, and Evaluator). Evaluating CoMind on MLE-Bench (or MLE-Live) and ongoing Kaggle competitions, the authors present that it outperforms the baselines in most settings.

**Strengths:**

1. Novelty of the main idea
The idea of incorporating real-time information from external communities for machine learning engineering agents is somewhat novel and interesting. In addition to machine learning competitions, CoMind has the potential to be leveraged for performing collaborative engineering tasks where human inputs are provided as real-time guidance, and this could be a more important potential application of CoMind.

2. Presentation
Overall, the manuscript is easy to follow and understand, primarily due to the fact that it contains intuitive figures as well as clear tables. Also, the authors show fair details of the experimental results, including statistics, prompts used, and examples of task completion.

**Weaknesses:**

1. Comparison across MLE-Live and MLE-Bench
While it is an interesting idea to take inputs from external communities, the community artifacts may contain too many hints. Although I think it makes sense to utilize the information under fair settings (such as comparisons with other Kaggle participants), comparing CoMind with other baselines on MLE-Bench (Table 1) is unlikely to be an apples-to-apples comparison. I'm aware that there are the results with CoMind but without the resources, but presenting the CoMind results as the primary finding could be misleading.

2. Reliance on and potential vulnerability to information from external communities
On the other hand, it may also pose a concern that such information from external communities might harm the AI agents and their progress. One of such cases would be inaccurate information; even though the information is from humans, it is still possible that some information is inaccurate or noisy. Another possibility is getting attacks with specific intentions to hinder the participating automated research agents' progress, where such attacks may come from either humans or other AI models. Having defense against them may be important for the proposed approach to be properly launched for real-world competitions or tasks.

3. Minor issues
- I believe the other entities in the MLE-Bench leaderboard have the error range in addition to the scores, whereas the error range is not presented by the authors.

**Questions:**

1. How did CoMind's scores improve as it iterated during the "ongoing" Kaggle competitions?

---

> ### Author Response · Authors · 2025-11-21
>
> We thank the reviewer for the constructive feedback and for recognizing the novelty of incorporating real-time community information into MLE agents. We value the assessment that CoMind has potential for broader collaborative engineering tasks. Below, we address the specific concerns.
>
> > W1: Comparison across MLE-Live and MLE-Bench
>
> A1: We appreciate the reviewer's concern and agree that giving CoMind access to public artifacts while some baselines lack retrieval mechanisms is not a perfectly symmetric setting. This is an inherent limitation of using the existing MLE-Bench leaderboard, where re-running all agents with identical resource conditions would require prohibitive API and compute cost, and several baselines are not designed to handle external resources at all. Meanwhile, many recent works with retrieval capability (e.g., MLE-Star) also report results on MLE-Bench under heterogeneous resource settings.
>
> We acknowledge the reviewer's point and will clarify in the paper that the MLE-Bench comparison is not intended as a perfectly controlled backbone-and-resource-matched study, but rather as evidence that **the design of CoMind and the MLE-Live framework has strong practical potential**.
>
> > W2: Reliance on and potential vulnerability to information from external communities
>
> A2: This is an insightful point. We agree that reliance on external data introduces risks of noise like erroneous code. However, CoMind is inherently designed to mitigate these risks through three mechanisms:
>
> - Verification, Not Blind Trust: CoMind does not blindly trust community ideas. It reproduces the code and evaluates it against the leaderboard. The agent acts as a defense layer, running candidate solutions against a local validation set. If a community artifact (whether noisy or malicious) fails to compile or yields poor validation scores, it is discarded during the evolutionary process.
> - Synthesis over Copying: Our agent, CoMind, does not simply copy publicly available kernels. The Idea Proposer is explicitly designed to brainstorm novel approaches that differ from the existing community codebase. This means the system is actively exploring alternatives, not blindly inheriting every suggestion.
> - Diverse Artifact Inclusion: As noted in the paper, MLE-Live includes the full publicly available corpus of community artifacts for agents to review, allowing the agent to perform cross-validation.
>
>
> > W3: I believe the other entities in the MLE-Bench leaderboard have the error range in addition to the scores, whereas the error range is not presented by the authors.
>
> A3: We acknowledge standard best practices regarding error bars. However, running the full MLE-Bench suite (75 competitions) is extremely resource-intensive, requiring substantial API cost and significant GPU hours per run. To compensate for this, we prioritized breadth and realism by evaluating on the “Ongoing” competitions, which offer a different form of robustness check beyond the static benchmark.
>
> > Q1: How did CoMind's scores improve as it iterated during the "ongoing" Kaggle competitions?
>
> A4: Since Kaggle imposes strict daily submission limits (preventing us from using the public leaderboard for every step), we tracked improvement using the Evaluator’s local validation score. We have revised the paper to include **Figure 7**, which illustrates the trajectory of CoMind's performance over time.

---

### Author Response · Authors · 2025-11-21

Thank all reviewers for their insightful and constructive comments! We are encouraged to learn that you recognize the novelty and significance of incorporating real-time community knowledge into MLE agents, as well as the value of our community-driven evaluation setup. We appreciate the positive feedback on the clarity and organization of our system design, the realism of the MLE-Live framework, and the strong empirical results on both retrospective and live Kaggle tasks. We are also glad that the ablations, qualitative analyses, and released code were found useful and informative.

We also appreciate the invaluable suggestions that help us further improve the paper. In response, we have made the following main revisions:

- Lack of clear data statistics (Reviewer 41Ga): We have added **Table 5** to provide comprehensive dataset statistics and clarify the scope of all evaluation resources.
- Missing detailed cost analysis (Reviewer 4mfQ and d6GG): We have added a new section describing CoMind’s API cost. A detailed breakdown of token usage is included in **Appendix C**.
- Additional clarification and analyses: We have added **Figure 7** illustrating CoMind’s score-improvement trajectory on ongoing Kaggle competitions (Reviewer d6GG) and **Appendix G** containing detailed error analysis (Reviewer 41Ga).

We have revised the paper accordingly, with all changes marked in red.

---

### Author Response · Authors · 2025-12-04

Dear AC,

Thank you for your time and effort in reviewing our submission. We understand that due to the recent OpenReview incident, reviewer scores were reverted to their pre-discussion values. To assist your assessment, we would like to summarize the review status as of November 25, prior to the system incident on November 27:

- Reviewer d6GG: Maintained score of 6; no discussion.

- Reviewer 41Ga: Score increased from 2 → 6 after our rebuttal (The reviewer explicitly confirmed the score raise during discussion.)

- Reviewer vka8: Maintained score of 6; no discussion.

- Reviewer 4mfQ: Maintained score of 6; the reviewer requested follow-up experiments to test transferability. Although the system incident stopped further exchanges, we subsequently ran experiments on CoBench, a combinatorial-optimization benchmark outside the Kaggle workflow, and found that CoMind outperforms strong baselines such as FunSearch. We believe that these results could directly address the reviewer's request for evidence of transferability beyond Kaggle-style settings.

- Overall: average score of 6 by Nov. 25th.

These score updates and discussions were all explicitly stated by the reviewers during the discussion period, and are preserved in the comment threads. We hope this clarification is helpful as you consider the final evaluation.

Thank you again for your consideration.

Sincerely,

The Authors

---

### Meta-Review · Area_Chair_2m46 · 2026-01-02

**Summary:**

This paper proposes a live framework to evaluate the ML agents in the community. The proposed CoMind includes five different agents to integrate public discussions and codes. The reviewers all believe that the paper is novel and interesting. Three reviewers gave positive scores and one reviewer is satisfied with the rebuttal. The authors addressed the concerns including enhancing the clarity on evaluation metrics and tasks. They also included the experiments on CoBench raised by reviewers.

Given this evidence and the authors' strong efforts, I'm happy to recommend acceptance.

**Reviewer Concerns:**

See above.

**Reviewer Scores:**

I believe one reviewer with a negative score will increase the score.

---

### Decision · Program_Chairs · 2026-01-26

Accept (Poster)